# Three-dimensional total-internal reflection fluorescence nanoscopy with nanometric axial resolution by photometric localization of single molecules

Alan M. Szalai [1], Bruno Siarry[1], Jerónimo Lukin[2], David J. Williamson [3], Nicolás Unsain[4], Alfredo Cáceres[4,5], Mauricio Pilo-Pais[6], Guillermo Acuna [6], Damián Refojo[2], Dylan M. Owen [3,7], Sabrina Simoncelli [3,8 ✉] & Fernando D. Stefani [1,9 ✉]

Single-molecule localization microscopy enables far-field imaging with lateral resolution in the range of 10 to 20 nanometres, exploiting the fact that the centre position of a single-molecule's image can be determined with much higher accuracy than the size of that image itself. However, attaining the same level of resolution in the axial (third) dimension remains challenging. Here, we present Supercritical Illumination Microscopy Photometric z-Localization with Enhanced Resolution (SIMPLER), a photometric method to decode the axial position of single molecules in a total internal reflection fluorescence microscope. SIMPLER requires no hardware modification whatsoever to a conventional total internal reflection fluorescence microscope and complements any 2D single-molecule localization microscopy method to deliver 3D images with nearly isotropic nanometric resolution. Performance examples include SIMPLER-direct stochastic optical reconstruction microscopy images of the nuclear pore complex with sub-20 nm axial localization precision and visualization of microtubule cross-sections through SIMPLER-DNA points accumulation for imaging in nanoscale topography with sub-10 nm axial localization precision.

[1] Centro de Investigaciones en Bionanociencias (CIBION), Consejo Nacional de Investigaciones Científicas y Técnicas (CONICET), Godoy Cruz 2390, C1425FQD Ciudad Autónoma de Buenos Aires, Argentina. [2] Instituto de Investigación en Biomedicina de Buenos Aires (IBioBA)-CONICET-Partner Institute of the Max Planck Society, Godoy Cruz 2390, C1425FQD Ciudad Autónoma de Buenos Aires, Argentina. [3] Department of Physics and Randall Centre for Cell and Molecular Biophysics, King's College London, London, UK. [4] Instituto Investigación Médica Mercedes y Martín Ferreyra-INIMEC-CONICET-Universidad Nacional Córdoba, Friuli 2434, X5016NST Córdoba, Argentina. [5] Instituto Universitario Ciencias Biomédicas de Córdoba (IUCBC), Friuli 2786, X5016NSW Córdoba, Argentina. [6] Department of Physics, University of Fribourg, Chemin du Musée 3, Fribourg CH-1700, Switzerland. [7] Department of Mathematics, Institute of Immunology and Immunotherapy and Centre for Membrane Proteins and Receptors, University of Birmingham, Birmingham, UK. [8] London Centre for Nanotechnology and Department of Chemistry, University College London, London WC1H 0AH, UK. [9] Departamento de Física, Facultad de Ciencias Exactas y Naturales, Universidad de Buenos Aires, Güiraldes 2620, C1428EHA Ciudad Autónoma de Buenos Aires, Argentina. ✉email: s.simoncelli@ucl.ac.uk ; fernando.stefani@df.uba.ar

maging the three-dimensional organization of biological structures down to the size of their structural proteins, ~4–10 nm, can open up exciting opportunities in the life sciences. Super-resolution microscopy, also known as far-field fluorescence nanoscopy, has set the conceptual pathway to achieve this goal[1–6]. Whereas in theory all super-resolution methods are able to reach nanometric resolution given a sufficiently high fluorescence photons flux, in practice most methods reach a lateral resolution limit of 10–20 nm. Axial resolution of methods using a single objective lens is typically two- to fivefold worse[7,8], including recent advances considering the experimentally determined microscope point-spread-functions[9], intensity-based approaches that rely on super-critical angle fluorescence[10] or photometric analysis of the defocused images of single molecules[11]. By exploiting the 4-Pi configuration[12] it is possible to reach an axial resolution below 35 nm, but at the cost of increased technical complexity. Isotropic stimulated emission depletion microscopy has been shown to deliver nearly isotropic resolution in the range of 30–40 nm[13,14], whereas 4-Pi photo-activated localization microscopy/stochastic optical reconstruction microscopy (STORM) has reached 10–20-nm resolution in 3D[15–17]. To date, sub-10-nm axial localization of single molecules was only achieved by two methods, metal-induced energy transfer (MIET) and MINFLUX. MIET decodes the $z$-position of fluorophores through lifetime imaging making use of the distance-dependent energy transfer from excited fluorophores to a metal film[18] or a graphene sheet[19,20]. However, combining this nanosecond time-resolved method with other nanoscopy methods in order to obtain 3D imaging with sub-10-nm resolution is not straightforward[21]. More recently, MINFLUX[22] was demonstrated to deliver sub-10-nm resolution in three dimensions[23], but this is at the cost of elevated technical complexity.

The use of the evanescent illumination field of total internal reflection to obtain sub-diffraction axial information in optical microscopy goes back to the 1950s and 1960s[24,25]. Total internal reflection fluorescence (TIRF) microscopy was pioneered by Axelrod in the early 1980s, demonstrating various applications including a scheme to obtain sub-diffraction axial resolution analysing the TIRF intensity as a function of the incidence angle[26]. These initial approaches were based exclusively on the axial dependency of the excitation intensity. Lanni et al. were the first ones to obtain axial positions from photometric readings of TIRF-illuminated 3T3 fibroblast cells excited at two different angles of incidence, and a theoretical calibration based on the model of Lukosz[27–29]. They could estimate average cell-substrate separation distances of 49 nm for focal contacts, and of 69 nm for close contacts[30]. In 1987 Axelrod revisited the work of Lukosz in the context of TIRF microscopy[31], and from then on, the knowledge to obtain axial positions from a quantitative use of TIRF microscopy was fully available.

Here, we introduce an easy-to-implement photometric method named Supercritical Illumination Microscopy Photometric z-Localization with Enhanced Resolution (SIMPLER) to determine the axial position of single molecules in a TIRF microscope. SIMPLER is able to deliver an axial localization accuracy comparable to MIET or MINFLUX, and at the same time requires no hardware modifications whatsoever to a conventional TIRF microscope and is fully compatible with all single-molecule localization microscopy (SMLM) methods. We demonstrate the performance of SIMPLER in combination with DNA points accumulation for imaging in nanoscale topography (DNA-PAINT) and direct STORM (dSTORM), achieving axial localization precisions smaller than 10 and 20 nm, respectively, throughout an axial range of 250 nm.

molecules from TIRF measurements involve two parts, a calibration of the TIRF signal and a method to estimate the axial position. The calibration of the TIRF signal can be obtained theoretically or experimentally. Theoretical calibration of the TIRF signal implies knowing the excitation evanescent field and the angular emission pattern of molecules at different $z$-positions. Calculating the evanescent field is a relatively straightforward task. By contrast, calculations of the angular emission pattern of molecules as a function of the distance to the interface are usually only at hand for someone with expertise in optics. Presumably for this reason, most works attempted to obtain calibrations using different experimental approaches. Examples include scanning axially quantum dots or fluorescent beads with a piezo actuator[32–34], or analysis of the emission intensity of fluorescent species fixed at different separations from the substrate. For the latter strategy, fluorophores were attached to a convex lens[35,36], large spherical beads[37], tilted microtubules[38,39], a tilted glass coverslip[40,41] or a nanometric staircase structure[42]. Remarkably, in most of these works, the effect of the substrate-sample interface on the emission power and angular emission was not present in the conceptual discussion. We note that, although an experimental calibration of the TIRF signal could include this effect, it is necessary considering the effect of the extra interface used to hold the calibration fluorescent probes in place, which will influence the detected signal too. Only in exceptional cases this was taken into account by matching the refractive indices of the liquid and the holding material[37,42], but this approach has a shortcoming too because the liquid used for the calibration usually has a refractive index different from the real samples.

With the calibration data at hand, previous methods have obtained axial positions or relative distances between fluorophores in two different ways. The one method estimates relative positions of particles within the evanescent field from the ratio of its fluorescence intensity under TIR and wide-field illumination[33,43,44]. The other method, usually called variable-angle TIRF, obtains absolute $z$-positions from TIRF measurements of the same sub-diffraction object at two[30] or more incidence angles[39,45–50], and fitting the intensity vs. incidence angle according to the calibration model. In order to determine the axial position of an object, both types of methods require sequential measurements of its emission under different illumination conditions. This is hardly compatible with the fast single-molecule blinking required for SMLM. Two recent works clearly demonstrate this limitation. On the one hand, Jung et al. made correlative measurements of cell membrane topography using variable-angle TIRF microscopy and SMLM. They could determine the membrane topography with 10–20-nm axial resolution but with diffraction-limited lateral resolution, and vice-versa for T-cell receptors[51]. On the other, Fu et al.[52] applied variable-angle TIRF to perform an effective optical sectioning with an axial resolution of 20 nm; each section was defined as the difference in the imaging depths obtained at the two adjacent angles of incidence. In each section, they performed SMLM and obtained a lateral resolution of about 100 nm.

By contrast, SIMPLER decodes $z$ information directly from 2D SMLM data. Through a full theoretical modelling, including the evanescent illumination, the modulation of the angular emission and the shape of the single-molecule signals in the image plane, we demonstrate that the TIRF intensity signal can be effectively represented by just three parameters, which are easily accessible. This parameterization allows the determination of the axial position of individual molecules from a single measurement of their emission intensity, which in turn enables the direct combination of SIMPLER with any SMLM method to obtain 3D super-resolved images.

## Results

**Principle of SIMPLER and theoretical axial localization precision.** Basically, all methods to obtain axial positions of

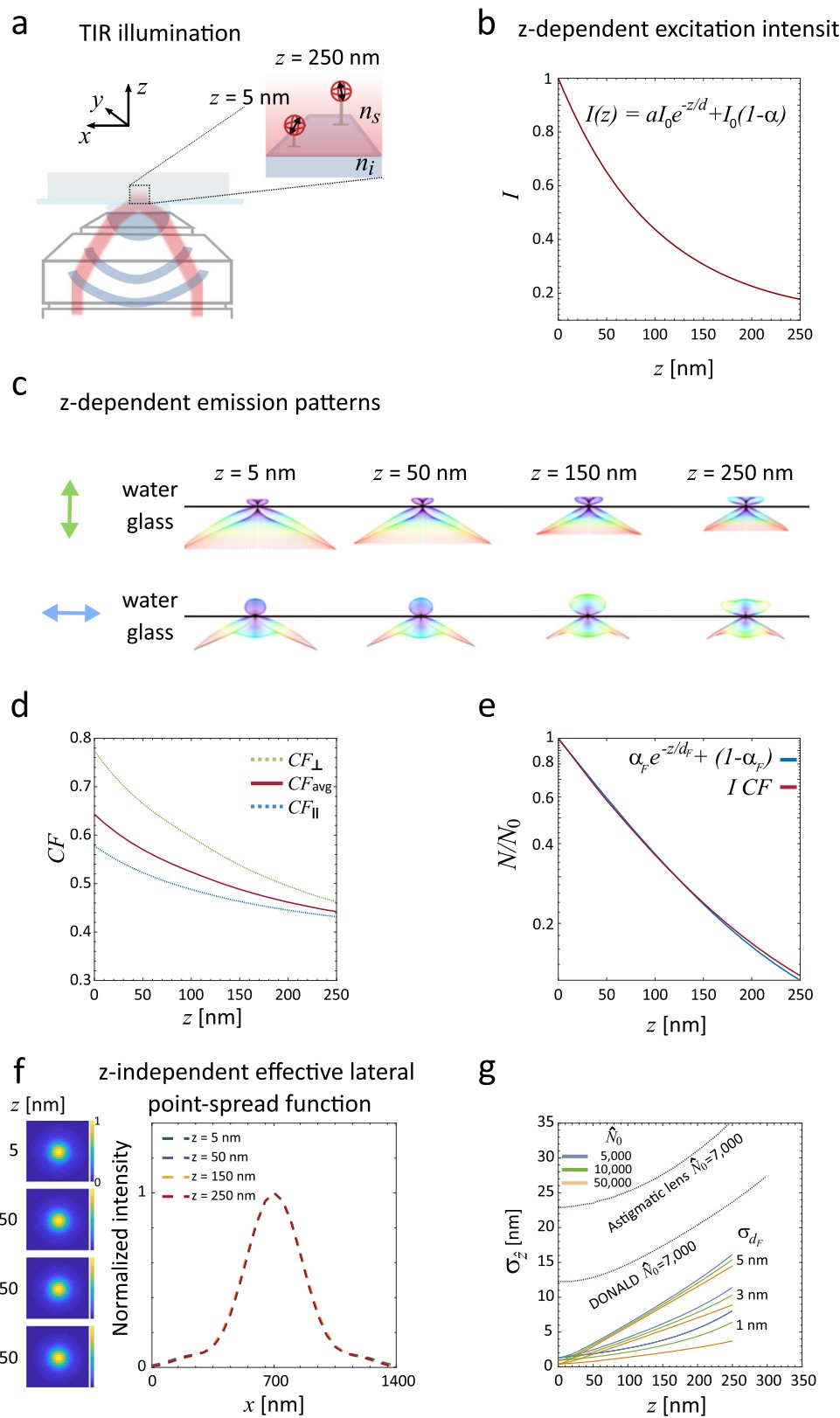

Figure 1 illustrates the concept of SIMPLER. Total internal reflection occurs when light impinges from a medium with refractive index $n_i$ on an interface with another medium of smaller refractive index $n_s < n_i$. If the angle of incidence $\theta_i$ is larger than the critical angle $\theta_C = \arcsin(n_s/n_i)$, light is fully reflected at the interface and an evanescent field appears,

penetrating the medium of low refractive index with an intensity that decays exponentially. In a fluorescence microscope, TIR illumination can be generated by controlling the angle of incidence of the excitation light using an immersion objective lens as schematically shown in Fig. 1a. In practice, the excitation field also contains a non-evanescent component due to scattering

**Fig. 1 Supercritical illumination microscopy enables photometric localization-enhanced z-resolution (SIMPLER). a** Simplified optical layout of the excitation path of a TIRF microscope exemplifying isotropic emitters, which corresponds to the case of the dipolar emitter rotating faster than the measurement time. **b** Intensity of the excitation field under TIR illumination for our experimental configuration: $\lambda_O = 642$ nm, $n_i = 1.517$, $n_s = 1.33$ water, $\theta_i = 69.5°$, $\alpha = 0.9$. **c** Simulated angular emission patterns of a dipolar emitter oriented either perpendicular (up) or parallel (bottom) to the water–glass interface and located at 5, 50, 150 and 250 nm above it. These calculations are made for $\lambda_O = 670$, the maximum emission wavelength of the fluorophores. **d** Fraction of fluorescence signal collected (CF collected fluorescence) with the microscope objective (NA = 1.42) for a fluorophore emitting at 670 nm and oriented parallel (blue dotted line) or perpendicular (green dotted line) to the glass/water interface, normalized to the case of a molecule far from the interface. The solid red line represents the isotropic average corresponding to the case of a rotating fluorophore. **e** Calculated z-dependent fluorescence signal (represented in logarithmic scale) of single molecules expressed as the ratio of number of photons detected at a given z-position ($N$) and the number of photons of an identical emitter placed at $z = 0$ ($N_O$). Principle of SIMPLER: the axial position of single molecules is retrieved from $N/N_O$ either through the exact solution ($I(z) \times CF_{avg}(z)$, solid red line) or through its exponential approximation (solid blue line—Eq. (2)). **f** Calculated images of single molecules at $z = 5$, 50, 150 and 250 nm with normalized intensity; xy images at the focal plane (right) and profiles along x (left). **g** Theoretical lower bound for the axial localization precision of SIMPLER (Eq. (3)) for different sets of $N_O$ and $\sigma_{dF}$ with $d_F = 87.5$ nm and $\alpha_F = 0.93$. Comparison of the theoretical localization precision of SIMPLER with respect to the reported precision of two other well-established z-localization techniques: single lens astigmatism and DONALD, for $\hat{N}_O = 7000$ photons (data taken from[10]).

in components of the optical system, that decays on a much longer scale[37]. Near the interface, the non-evanescent component can be considered constant and the overall illumination field is represented by a linear superposition of both contributions, $I(z) = \alpha I_0 e^{-z/d} + (1-\alpha)I_0$ with $I_0$ the intensity at the interface, $d = \lambda_0/4\pi/\left(n_i^2 \sin^2(\theta_i) - n_s^2\right)^{1/2}$ the penetration depth, $\lambda_0$ the vacuum wavelength and $1-\alpha$ the scattering contribution fraction. The excitation rate of a freely rotating fluorophore (under linear excitation) will depend on its axial position according to $I(z)$. Figure 1b shows $I(z)$ for one of our experimental configurations ($\lambda_0 = 642$ nm, $n_i = 1.517$, $n_s = 1.33$ water, $\theta_i = 69.5°$, $\alpha = 0.9$), which decays with $d = 102$ nm.

Within the range of TIRF, the process of fluorescence emission is also influenced by the dielectric interface[53]. Figure 1c shows that the calculated angular emission patterns of fluorophores oriented parallel and perpendicular to the glass–water interface, for four different axial positions within the penetration depth of the evanescent field ($z = 5$, 50, 150 and 250 nm). Clearly, for both emitter orientations, fluorophores emit more fluorescence into the glass semi-space as they get closer to the interface. The dotted curves in Fig. 1d are the integrals of the angular emission pattern over the collection solid angle of a microscope objective with numerical aperture NA = 1.42. These curves represent the collected fluorescence (CF) from single molecules oriented parallel and perpendicular to the interface, as a function of the axial position, and normalized to the case of a fluorophore far from the interface. In addition, the isotropic average ($CF_{avg}$) is also shown, which corresponds to the usual experimental situation in SMLM methods where fluorophores, bound to antibodies (i.e. dSTORM) or oligonucleotides (i.e. DNA-PAINT) via flexible bonds, can rotate during the measurement time.

Then, for a single molecule located in the evanescent field, the detected fluorescence signal will be proportional to the product of the excitation field and the CF: $F(z) = I(z) \times CF_{avg}(z)$ (hereafter referred to as the exact solution). It turns out that $F(z)$ is well-represented by an exponential function analogous to $I(z)$ but with a steeper decay ($d_F = 87.5$ nm) and smaller background constant ($\alpha_F = 0.93$). Figure 1e shows the exact solution (red curve) and the exponential approximation (blue curve). The difference between them is negligible (<1 nm for $z < 150$ nm, and <8 nm for $z < 250$ nm) (Supplementary Fig. 1a). This dependency of the fluorescence signal with the axial position is the core of SIMPLER axial localization.

In the context of SMLM, it is convenient to express the fluorescence signal in terms of the number of photons, $N$, detected in a given unit of time (typically the acquisition time of a camera frame)

$$N(z)/N_0 = \alpha_F e^{-z/d_F} + (1-\alpha_F) \qquad (1)$$

where $N_0$ is the number of photons emitted by a fluorophore at $z = 0$.

Also, relevant for SMLM is the fact that the variations of the angular emission pattern do not produce any significant modification of the shape of single-molecule signals in the image plane. Figure 1f shows the calculated single-molecule signals obtained by focusing the collected fractions (NA = 1.42) of the angular emission patterns of molecules at $z = 5$, 50, 150 and 250 nm, and normalized profiles. This means that $N(z)$ can be estimated with a single procedure throughout the TIRF range.

Following Eq. (1), an experimental estimation of the axial position of a molecule ($\hat{z}$) can be obtained from a measurement of the number photon counts detected in a camera frame time ($\hat{N}$), knowing the value of photon counts at $z = 0$ $\left(\hat{N_0}\right)$ for an identical emitter

$$\hat{z} = d_F \times \ln \frac{\alpha_F}{\hat{N}/\hat{N_0} - (1-\alpha_F)} \qquad (2)$$

Equation (2) expresses the working principle of SIMPLER. Based on it, we analyse the theoretical maximum achievable accuracy of SIMPLER for axial localization. The standard error of $\hat{z}$, which ultimately determines the axial resolution in SMLM, can be estimated as:

$$\sigma_{\hat{z}} = \sqrt{\left(-\frac{d_F}{\hat{N} - (1-\alpha_F)\hat{N_0}}\right)^2 \times \sigma_{\hat{N}}^2 + \left(\ln \frac{\alpha_F \hat{N_0}}{\hat{N} - (1-\alpha_F)\hat{N_0}}\right)^2 \times \sigma_{d_F}^2} \qquad (3)$$

This expression, while it only considers uncertainties in $\hat{N}$ ($\sigma_{\hat{N}}$) and $d_F$ ($\sigma_{dF}$), is useful to obtain a theoretical lower bound for $\sigma_{\hat{z}}$. As the theoretical lower bound for $\sigma_{\hat{N}}$, we considered it equal to $\sqrt{\hat{N}}$, which arises from the fact that $\hat{N}$ is Poisson-distributed and that in SMLM the photon counts of each fluorophore are typically determined in one single measurement. We note, however, that in real-life experiments, other factors may enlarge this value. For example, the variance introduced by EM amplification in EM-CCD cameras used in SMLM can lead to errors in photon counts that are a factor of 2 larger than Poisson statistics[54]. Also, the presence of a faster blinking process occurring with rates comparable to the camera frame rate may increase the variability of $\hat{N}$. Figure 1g displays curves of $\sigma_{\hat{z}}$ as a function of the axial position computed with Eq. (3), taking $\sigma_{\hat{N}} = \sqrt{\hat{N}}$ for different

experimentally accessible values of $N_0$ and $\sigma_{dF}$ ($d_F$ and $\alpha_F$ were fixed to 87.5 and 0.93 nm, respectively). $N_0$ was varied between 5000 and 50,000 photons per frame to account for a wide range of experimental conditions (different laser powers, frame rates, fluorophores, etc). On the other hand, $\sigma_{dF}$ was ranged from 1 to 5 nm as these values relate to sensible extreme limits for the standard error of $\theta_i$ (0.3°–1.5°). This shows that under usual experimental conditions, SIMPLER is potentially able to deliver axial localization precisions of just a few nanometres. Interestingly, a useful range with axial localization precision below 10 nm can be obtained, depending on the uncertainty of $d_F$. For $\sigma_{dF}$ 1 nm, the sub-10-nm axial localization precision range extends up to $z = 250$ nm, for $\widehat{N_0} > 10,000$. If $\sigma_{dF} = 5$ nm or larger, the axial localization precision becomes fairly independent of the photon count for $\widehat{N_0} > 30,000$, but the range of sub-10-nm axial localization precision is limited to $z < 170$ nm. For comparison, Fig. 1g also shows the performance of two other methods for axial localization. Within the TIR penetration depth, SIMPLER is in principle able to achieve superior performance than the commonly used single cylindrical lens configuration[7] and more recent approaches that decode axial position from fluorescence emission at supercritical angles (i.e. DONALD[10] and SALM[55]). At the same time, SIMPLER holds the additional advantage that it does not require any hardware modification whatsoever to a conventional SMLM TIRF microscope, and that data are acquired and analysed in essentially the same way.

**Experimental implementation of SIMPLER**. We performed experiments to characterize SIMPLER in two different SMLM TIRF microscopes. A custom-built microscope and a commercial microscope with flat-illumination optics and internal calibration of the angle of incidence (Nikon N-STORM 5.0). Further details of each set-up are provided in 'Methods'.

The input data for SIMPLER is a list of single-molecule localizations, where in addition to the $xy$ location, the number of photons per frame $\left( \widehat{N} \right)$ (from which the $z$ coordinate will be determined through Eq. (2)) and the frame number information must be included (the reason for the latter will be clarified below). Most of the available software for SMLM image reconstructions already count with routines to output $\widehat{N}$ and frame number. Thus, the implementation of SIMPLER requires only a few modifications in the acquisition and analysis. The sequence of additional steps to implement SIMPLER in any SMLM TIRF experiment are summarised in the work-flow diagram presented in Supplementary Fig. 2 and discussed in detail below.

First, in order to assure an accurate determination of $\widehat{N}$, frames where it is uncertain whether the molecule was emitting during the complete integration time have to be discarded. This is the case of the first and last frame of a single-molecule emission event, as it can be seen in Fig. 2a. Thus, the acquisition conditions should be adapted to record the average single-molecule emission event during more than three camera frames, enabling an additional frame-filtering step during analysis to exclude the first and last frames of each single-molecule trace. This frame-filtering rules out low-intensity events that would bias axial localizations to artificially higher $z$ values. In the case of DNA-PAINT measurements, the frame filtering has the additional advantage of excluding short emission events due to non-specific binding of the labelled oligonucleotides.

The excitation light intensity may not be uniform throughout the complete field of view of a wide-field microscope. For example, in our custom-made microscope, wide-field illumination is achieved using the central part of an expanded (nearly Gaussian) beam. Alternatively, some microscopes include special optics, such as apodizing neutral filters, to attain a flat(ter) illumination profile. That is the case of the commercial instrument N-STORM 5.0. SIMPLER works in both kinds of set-up. In the case of non-uniform illumination, the photon count of each molecule needs to be corrected using the local background level as a measure proportional to the local excitation intensity (see 'Methods' for details about the correction procedure).

Determination of $z$ coordinates through Eq. (2) requires previous knowledge of $N_0$, $d_F$ and $\alpha_F$. Next, we explain how these parameters are obtained.

$N_0$ is the number of detected fluorescence photons for fluorophores located at $z = 0$. It can be determined in SMLM measurements by computing the average $\widehat{N}$ for fluorophores bound to structures whose distance to the coverslip is negligible. Such structures could be present in the same biological sample (e.g. if a known cellular component is known to be attached or very close to the substrate), or in another sample made specially to obtain $N_0$. In the latter case, we found it practical to deposit directly on a coverslip the same fluorescent labels used for biological imaging and determine their average $\widehat{N}$ under identical experimental conditions as in the biological imaging experiments. For example, Fig. 2b shows the histogram photon counts per frame used to determine $\widehat{N_0}$ for a SIMPLER–DNA-PAINT measurement. It was obtained with a sample where the same secondary antibody fragments (Fab) used for labelling the biological samples were deposited on a coverslip and imaged through DNA-PAINT under identical conditions with the complementary fluorescently labelled DNA imager strand (Fig. 2b, inset). The distribution of $\widehat{N_0}$ (after the correction for local intensity and frame filtering) is well-described by a normal distribution with an average value of 51,000 photons and a standard deviation of 10% (for comparison, Supplementary Fig. 3 shows the non-frame filtered photon count histogram). Such a determination of $\widehat{N_0}$ is sufficiently accurate for SIMPLER, as shown in the performance examples of the next section where the 3D nanoscale organization of different biological structures is resolved. Specifically, we have quantified the effect of using SIMPLER with a wrong value of $\widehat{N_0}$ in Supplementary Fig. 1d. It turns out that applying SIMPLER with the wrong $\widehat{N_0}$ produces mainly a localization off-set for molecules located at $z$ between 0 and 150 nm. For molecules located at larger $z$, small axial distortions are introduced. For example, an overestimation of $\widehat{N_0}$ by 10% leads to a mislocalization $\Delta z = 8.5$ nm for molecules at $z = 0$. For molecules at $z = 150$ nm, $\Delta z$ increases to 11.0 nm, and for molecules at $z = 250$ nm $\Delta z$ is 18.1 nm. A similar behaviour with negative mislocalizations is found for an underestimation of $\widehat{N_0}$ by 10%.

$d_F$ and $\alpha_F$ are obtained from a fit to the expected $z$-dependent single-molecule fluorescence intensity $F(z) = I(z) \times \text{CF}_{\text{avg}}(z)$ (Fig. 1e). The evanescent component of $I(z)$ is readily obtained from the decay constant $d = \lambda_0 / 4\pi / \left( n_i^2 \sin^2(\theta_i) - n_s^2 \right)^{1/2}$, which depends on available experimental parameters. In the custom-built microscope, the angle of incidence $\theta_i$ was determined with a 0.7° uncertainty by analysing the lateral displacement of the focus as a function of axial displacement, as described in ref. [42] (see Supplementary Fig. 4 and 'Methods'). In the commercial microscope, we relied on the internal calibration of incidence angle. Both ways of obtaining $\theta_i$ delivered satisfactory results. Determining accurately the relative contribution of the non-evanescent field $(1 - \alpha)$ can be challenging, as recently reviewed[56]. In objective-type TIRF microscopes, $(1 - \alpha)$ is typically found to be between 10 and 15%[42,56]. Fortunately, SIMPLER is rather insensitive to variations of $\alpha$ in that range. For example, a

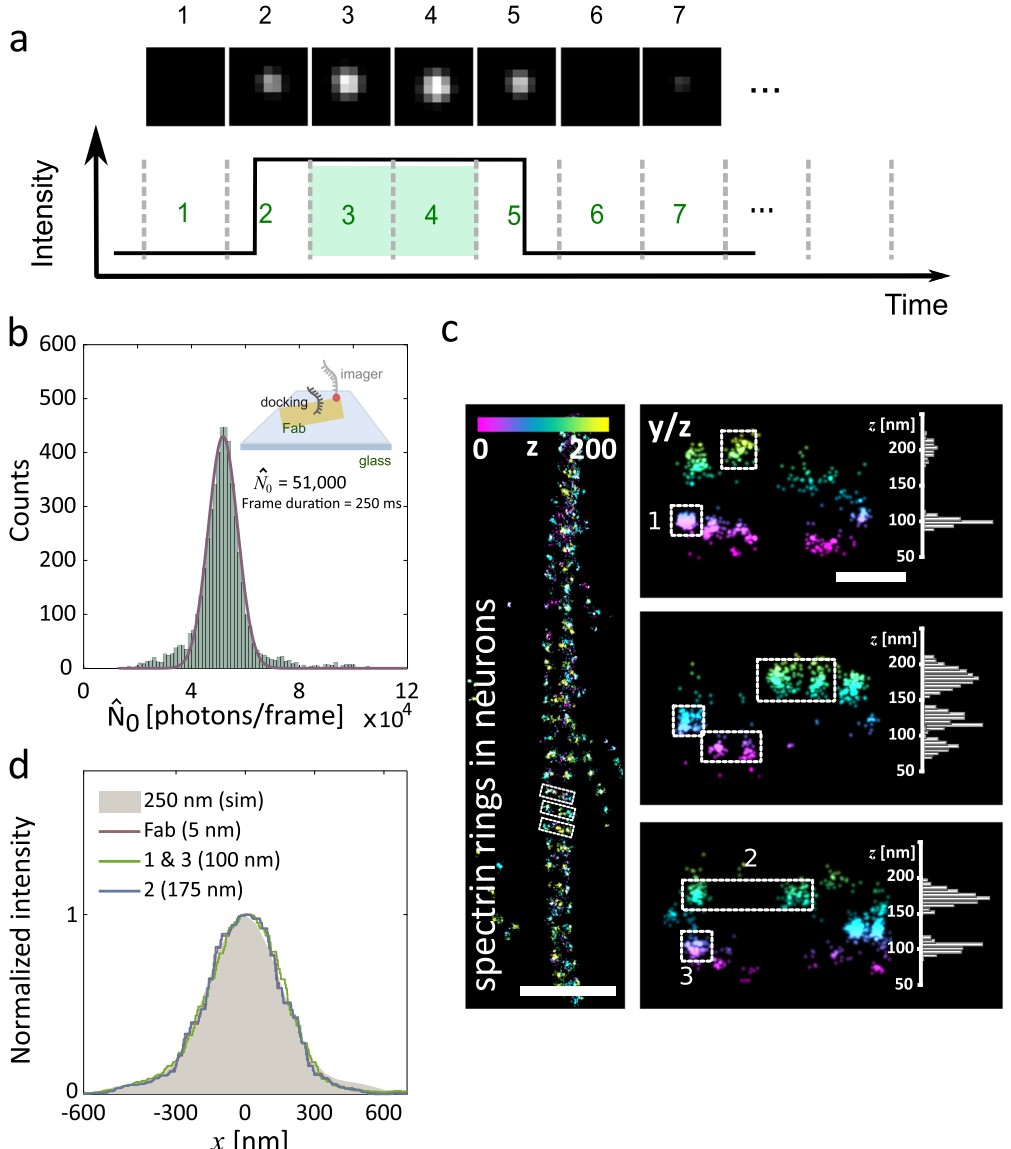

**Fig. 2 Data analysis and z assessment via SIMPLER. a** Single molecules are detected and localized using any single-molecule localization microscopy software. In the case of non-uniform illumination experiments (i.e. due to Gaussian shape of the excitation beam), the number of photons of each localization is corrected to the local excitation intensity (see 'Methods'). Next, a frame-filtering step is performed to only use localizations that lasted at least three frames and compute their photon count per frame excluding potentially misleading first and last frames. **b** Histograms of photons counts for ATTO655, imaged using DNA-PAINT, with a sample of DNA-labelled (docking strand) Fab fragments deposited over a coverslip. The imaging area was $16 \times 16 \ \mu m^2$ with a 4-Hz acquisition frame rate (see 'Methods'). Inset: schematic representation of the experimental conditions. **c** β2-spectrin rings in hippocampal neurons. Left: top view ($xy$). Right: magnified side-views ($yz$) of the boxed regions in the top view, together with axial profiles of the boxed areas. Localizations are colour-coded according to their $z$-position. Top view and side-views have the same $z$-colour scale. The ring-like structure shown was similarly reproduced in several axons from three independent experiments. **d** Normalized image profiles of single emitters located at different axial positions ($z = 5$, 100 and 175 nm) within the TIRF region along with the simulated profile for a rotating dipole located at $z = 250$ nm. Experimental profiles were obtained from samples of Fab fragments adsorbed to the coverslip ($z = 5$ nm) and β2-spectrin rings in hippocampal neurons ($z = 100$ and 175 nm, numbered regions in **c**). Scale bars represent 1 μm (**c**, top view); and 100 nm (**c**, side view). Number of localizations kept after frame-filtering step: (**b**) 5030 (out of 21,639); (**d**) (top) 625 (out of 1215), (centre) 517 (out of 1157) and (bottom) 892 (out of 1563).

misleading value of $(1 - \alpha)$ in the range of 8–12% introduces a maximum accumulated distortion smaller than 5 nm in the $z$ range from 0 to 150 nm. This means that a structure occupying the whole axial range of 0–150 nm, would be imaged with an accumulated distortion smaller than 5 nm. For a larger structure, occupying the range from 0 to 250 nm, the total accumulated axial distortion would be of about 10 nm (Supplementary Fig. 1c). It should be noted that these are total accumulated distortions smaller than 4%, hardly noticeable and probably surpassed by other factors such as the size or the position of the labels. Overall,

the robustness of SIMPLER against incorrect values of the calibration parameters $\theta_i$, $\alpha$ and $N_0$, is described in Supplementary Table 1 and Supplementary Figs. 1, 5, 6.

Obtaining $CF_{avg}(z)$ requires the calculation of the angular emission pattern of single molecules and their integration over the NA of the objective. Details about these calculations are provided in 'Methods'. In order to simplify this task for new users of SIMPLER, in the Supporting Information we provide tabulated values of $CF_{avg}(z)$ for the most usual TIRF microscopy configurations, namely NA ranging from 1.40 to 1.49 and

fluorophore maximum emission wavelength ($\lambda$) between 500 and 720 nm (Supplementary Table 2).

To ease the implementation of SIMPLER to the wide-imaging community, we also make available a Supplementary Software with an intuitive graphical user interface that directly outputs $z$-position from 2D SMLM analysis lists (Supplementary Software). The software computes $F(z) = I(z) \times CF_{avg}(z)$ and provides $d_F$ and $\alpha_F$ for each user input experimental conditions: NA; $\lambda_0$; $\lambda$; $\theta_i$; $n_i$; $n_s$ and $\alpha$. Among other features, the software can correct photon counts for uneven illumination and automatically perform the frame-filtering step. In addition, by taking advantage of the distinctive effects of each of the calibration parameters $\theta_i$, $\alpha$ and $N_0$ (Supplementary Table 1 and Supplementary Figs. 1, 5, 6), their values can be easily adjusted using 3D images of reference structures as feedback. The Supplementary Software includes a specific module for this purpose and, as illustrated in Supplementary Fig. 7, it is extremely useful to find the best estimate for a parameter that has been determined or estimated with low accuracy.

**3D SIMPLER SMLM of biological samples**. We reconstructed super-resolved images by plotting all valid localizations after frame-filtering (further details in 'Methods'). Figure 2c shows an example of SIMPLER–DNA-PAINT to deliver 3D images of the regular arrangement of β2-spectrin in the membrane-associated periodic skeleton (MPS) of neuronal axons. Whereas the top view image in Figure 2c clearly shows the MPS characteristic period of 190 nm, the 3D imaging using SIMPLER allows resolving the sub-membrane organization of β2-spectrin across the axon and to identify single-molecule signals corresponding to spectrin molecules positioned at different heights. We used these images to corroborate the predicted (Fig. 1f) invariability of the effective lateral point-spread function within the usable TIRF depth. Figure 2d shows the normalized profiles of average signals obtained at 5 (Fabs deposited on the glass substrate), 100 or 175 nm (spectrin), and the calculated signal at 250 nm obtained from focusing the calculated emission pattern. When normalized, all signals were indistinguishable, independently of their axial position, confirming that a single algorithm can be used to obtain the photon counts of molecules positioned throughout the TIRF range. In addition, we used this biological structure to compare 3D SIMPLER performance when the first and last frames of each single-molecule trace are ruled out. In Supplementary Fig. 8, it can be seen that different clusters are enlarged in the axial direction toward higher $z$ values when those frames are unfiltered, thus confirming the importance of performing this post-processing step.

Figures 3 and 4 further illustrate the performance achievable with SIMPLER SMLM to deliver super-resolved 3D images of biological structures. Figure 3 shows a 3D image of the microtubule network of COS-7 cells imaged through DNA-PAINT. Figure 3a includes a top ($xy$) view of the microtubule network alongside with four cross-sectional views of individual microtubules. SIMPLER can fully resolve the hollow circular structure of immunolabeled microtubules, one of the smallest supramolecular protein structures in biological cells. Fitting a circle to 50 cross-sections of microtubules retrieves an average diameter of 41 nm with a standard deviation of 6 nm (Fig. 3b), in good agreement with what is expected for an immunolabeled microtubule (primary antibodies and Fab fragments from secondary antibodies)[57]. In comparison to other methods that have achieved this level of axial resolution[9,17], all of which involve high technical complexity, SIMPLER delivers equivalent or better resolution using the hardware of a conventional TIRF microscope with no modifications.

Imaging with this level of resolution in 3D makes it possible to resolve bundles of microtubules that we found to be usual in hippocampal neurons, and that would otherwise be interpreted as single microtubules (Supplementary Fig. 9). The hollow cross-sections of individual microtubules were also resolved through SIMPLER–DNA-PAINT images of microtubules in Human Fetal Foreskin Fibroblasts cells, acquired in a commercial TIRF microscope (Nikon N-STORM 5.0). In this case, we used the instrument internal calibration of the incidence angle, and applied SIMPLER directly, using the calculated parameters for that system and without even correcting for non-uniform illumination (the microscope was equipped with a gradient neutral-density filter to produce a near-flat-top intensity profile from the Gaussian-shaped beam input). These results (Supplementary Fig. 10) demonstrate that SIMPLER can be applied to SMLM data acquired with commercial set-ups, and that can be easily adopted by any user.

To obtain insight into the critical parameters for SIMPLER, we determined experimentally the axial localization precision ($\sigma_{\widehat{z}}$) and compared it to the theoretical predictions (Eq. 3). From the same SIMPLER–DNA-PAINT experiment ($\widehat{N}_0 = 51{,}000$), single-molecule emission events longer than 5 camera frames were selected to compute an experimental measure of $\sigma_{\widehat{z}}$. In this way, after filtering out the first and last frames (Fig. 2a), at least three independent measurements of $\widehat{N}$, and their corresponding estimations of $\widehat{z}$, were available for every single molecule. Figure 3c shows the obtained distributions of experimental $\sigma_{\widehat{z}}$ (extracted from 366 single-molecule traces) grouped for different ranges of $z$. The average axial localization precision is well below 10 nm throughout the complete $z$ range. This observation was also supported by image-based decorrelation analysis[58] of the microtubules cross-sections presented in Fig. 3a, resulting in an estimated average spatial resolution of ~11 nm (Supplementary Fig. 11a).

The theoretical lower bounds of $\sigma_{\widehat{z}}$ shown in Fig. 1g were calculated using $\sigma_{\widehat{N}} = \sqrt{\widehat{N}}$, i.e., considering $\widehat{N}$ to be Poisson-distributed. From the single-molecule traces, we have determined the experimental variance of $\widehat{N}$ (Fig. 3d), which presents an average value of around $5\sqrt{\widehat{N}}$. With this data at hand, we can make a direct comparison of the experimental performance of SIMPLER to the theoretical predictions. In Fig. 3e, we plotted the median of the distributions of $\sigma_{\widehat{z}}$ shown in Fig. 3d against its $z$ median. For comparison, we plotted the theoretical curves calculated from Eq. (3) for a set of values of $\sigma_{\widehat{N}}$ and $\sigma_{dF}$. The experimental values of $\sigma_{\widehat{z}}$ agree well with the theoretical prediction with values of $\sigma_{dF}$ between 2 and 4 nm and $\sigma_{\widehat{N}} = 5\sqrt{\widehat{N}}$. We remark though that this analysis is only performed to validate the applicability of Eq. (3). It is not necessary for the application of SIMPLER, as its implementation does not require any prior knowledge about the variances of the parameters involved. On the other hand, it should also be noticed that, while the axial localization precision at any given point within the field of view is well-described by Eq. (3), if we compare the absolute axial position of different molecules over the entire field of view, the axial position variability will be higher. This is due to the position-dependent variability of $\widehat{N}_0$ throughout the field of view because the illumination is not perfectly uniform or corrected. To illustrate this, Supplementary Fig. 12 shows the field-dependent axial distribution obtained with SIMPLER–DNA-PAINT for a sample of DNA-Fab fragments adsorbed to the coverslip. We quantified that as the field of view increases from 25 to 256 μm², the overall axial variability increases from 7 to 16 nm. Taking into account the orientation and the size of the Fab

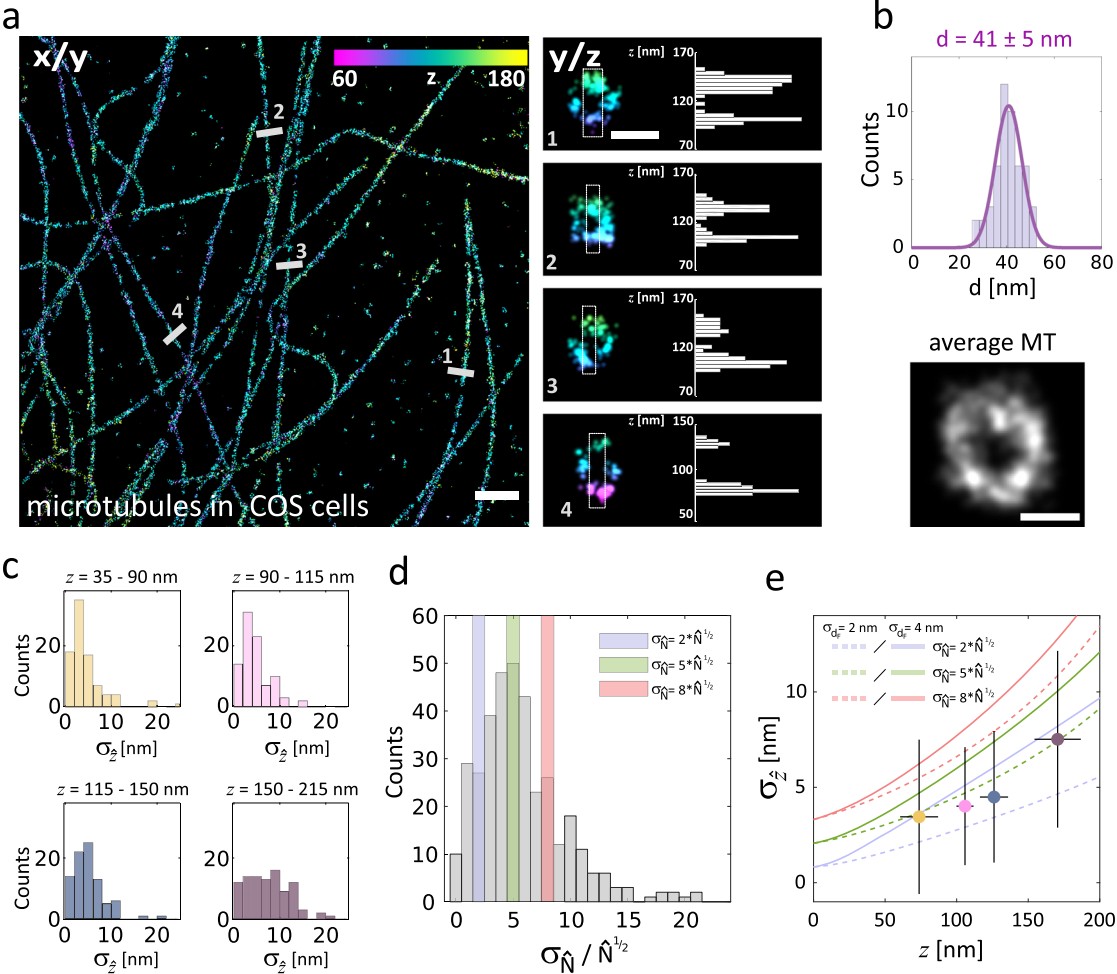

**Fig. 3 Microtubules immunolabeled for DNA-PAINT super-resolved in 3D using SIMPLER. a** Microtubules in COS-7 cells. Left: top view. Right: magnified side-views along the numbered lines in the top view, together with the axial profile of the boxed areas. Top- and side-views have the same $z$-colour scale. Three independent experiments were performed with similar results. **b** Distribution of microtubule diameters ($n = 50$), with an average of 41 nm and a standard deviation of 6 nm. An average ($n = 8$) microtubule profile is also shown. **c** Histograms of $\sigma_{\hat{z}}$ at different $z$-positions, obtained experimentally from 366 DNA-PAINT single-molecule traces. Data include traces from microtubules in COS-7 cells, as well as from spectrin in hippocampal neurons (Fig. 2d). **d** Histogram of the experimentally determined variance of $\hat{N}$ ($\sigma_{\hat{N}}$), expressed in units of $\sqrt{N}$ to remark how much larger is $\sigma_{\hat{N}}$ with respect to the expected theoretical lower bound ($\sigma_{\hat{N}} = \sqrt{N}$). The values 2, 5 and 8 are highlighted. The variance was obtained from the same 366 single-molecule traces as those analysed in **c**. **e** Median $\sigma_{\hat{z}}$ values located in the median $z$-position of each interval defined in **c**, overlapped with theoretical curves of $\sigma_{\hat{z}}$ for $\hat{N}_O = 51,000$ and $\sigma_{\hat{N}} = 2x$, $5x$ or $8x\sqrt{N}$ for $\sigma_{dF} = 2$ and 4 nm. Error bars represent the standard deviation of $\sigma_{\hat{z}}$ and $z$. The number of single-molecule traces analysed in each interval was 93 (for $z = 35$–90 nm); 90 (for $z = 90$-115 nm); 87 (for $z = 115$–150 nm); and 96 (for $z = 150$-225 nm). Data includes localizations from three biologically independent samples (immunolabeled microtubules in two independent COS-7 cells and immunolabeled spectrin in a hippocampal neuron). Scale bars represent 1 μm (**a**, top view); 50 nm (**a**, side view) and 25 nm (**b**). Number of localizations kept after frame-filtering step: **a** (1) 176 (out of 292), (2) 257 (out of 433), (3) 131 (out of 294) and (4) 85 (out of 275).

fragments ($z$-position of the fluorophores range from 0 to ~5 nm), as well as the field-dependent correction of the number of photons by the excitation profile, these broadening of the axial distributions are consistent with the individual localization precisions reported above.

In Fig. 4 we show another example of application of SIMPLER, in this case in combination with dSTORM, to obtain 3D super-resolved images of the stereotypical arrangement of the nucleoporin Nup107 in the nuclear pore complex of HeLa Kyoto cells. These experiments were carried out in a Nikon N-STORM 5.0 microscope. Figure 4a shows top-view images of a nucleus, where many nuclear pore complexes are visible. Even though the labelling efficiency was sub-optimal, the typical eightfold symmetry of the complex is evident in many cases. More

importantly, SIMPLER clearly resolves the axial separation of the cytoplasmic and nucleoplasmic rings. Figure 4b shows the lateral and axial cross-sections of an average nuclear pore complex. The axial separation distance between the cytoplasmic and nucleoplasmic rings is determined to be 59 nm (Fig. 4c), in excellent agreement with previous reports[59,60].

The SIMPLER–dSTORM data were acquired with $\widehat{N}_0 = 10,000$ photons. The experimentally determined axial localization precision (161 single-molecule traces) was found to be below 20 nm throughout the 0–250-nm axial range (Fig. 4d), while the spatial resolution of the x/z image shown in Figure 4b, determined through image decorrelation analysis[58] was ~34 nm (Supplementary Fig. 11b). With this level of axial localization precision, the cross-sections of single microtubules and bundles (which are

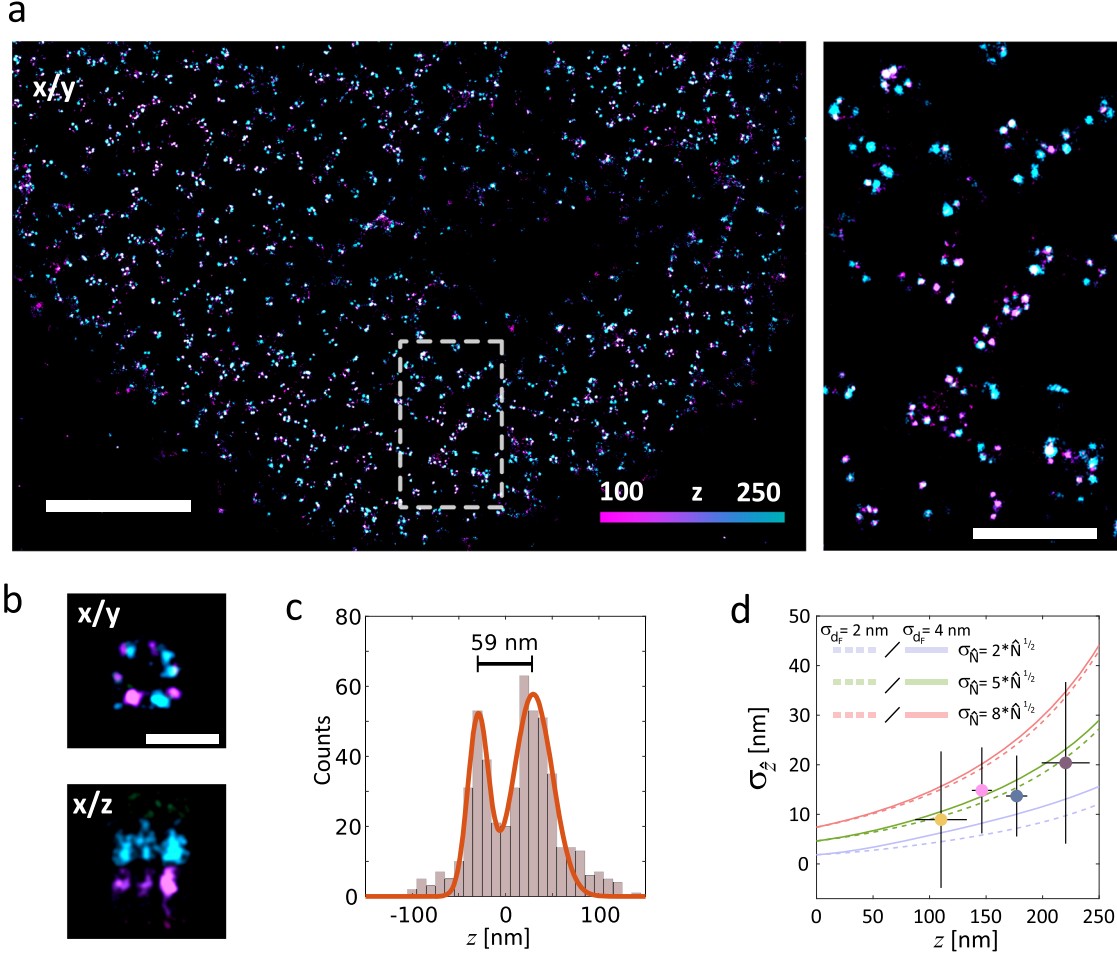

**Fig. 4 NUP107 immunolabeled for dSTORM and super-resolved in 3D using SIMPLER. a** Left: top view of dSTORM image of NUP107-mEGFP in HeLa Kyoto cells, labelled with primary anti-GFP antibody conjugated to Alexa Fluor 647. Right: magnified view of the region marked. Four independent experiments were performed with similar results. **b** Average nuclear pore complex showing the archetypal eightfold symmetry (*xy* top-view, top) and the organization in nuclear and cytoplasmic rings (*xz* side-view, bottom) reconstructed by SIMPLER ($n = 4$). Top- and side-views have the same *z*-colour scale as defined in **a**. **c** Histogram of *z*-positions of the nuclear pore complex shown in **b** yields 59-nm separation between the nuclear and cytoplasmic rings. **d** Median $\sigma_z$ values of different *z*-positions obtained experimentally from 161 dSTORM single-molecule traces overlapped with theoretical curves of $\sigma_z$ for $\hat{N}_O = 10{,}000$ and $\sigma_{\hat{N}} = 2x$, 5x or $8x\sqrt{\hat{N}}$ for $\sigma_{dF} = 2$ and 4 nm. Error bars represent the standard deviation of $\sigma_z$ and *z*. The number of single-molecule traces analysed in each interval was 59 (for $z = 35$–130 nm); 45 (for $z = 130$–160 nm); 30 (for $z = 160$–190 nm) and 23 (for $z = 190$–250 nm). Data include localizations from four biologically independent samples (four HeLa Kyoto cells from independently prepared samples). Scale bars represent 2 μm (**a**, left); 500 nm (**b**) and 100 nm (**c**). Number of localizations kept after frame-filtering step: **b** 1101 (out of 2889).

considerably smaller than the nuclear pores and thus far more challenging to visualize) are also visible. Supplementary Figure 13 shows example images of microtubule cross-sections obtained by 3D SIMPLER–dSTORM.

Figure 4d also shows that the experimentally determined $\sigma_z$ is in good agreement with the theoretical prediction (Eq. (3)) with $\sigma_{dF}$ between 2 and 4 nm and $\sigma_{\hat{N}} = 5\sqrt{\hat{N}}$. This result, together with the good agreement between calculated and experimental $\sigma_z$ both for SIMPLER–DNA-PAINT (Fig. 3e), indicates that the axial localization precision achievable with SIMPLER in combination with any other SMLM method can be predicted using Eq. (3).

## Discussion

We have presented and characterized, theoretically and experimentally, a photometric method to localize single fluorescent molecules with nanometric precision in the axial direction of a TIRF microscope. SIMPLER decodes the axial position (*z*) of single molecules based on three phenomena: the *z*-dependency of

the excitation intensity, the *z*-dependency of the angular emission, and the *z*-independent effective lateral point-spread function of the single-molecule signals in the image plane. Functional analysis of the *z*-dependent single-molecule intensity enables its calibration based on just three parameters, that are easily accessible or provided in this work for most usual experimental configurations. We provide a Supplementary Software to implement SIMPLER from a list of localizations as obtained from any SMLM experiment. The software has specific routines to perform all the required analysis for SIMPLER. Also, it can help users detect if any calibration parameter is inaccurately set, through a module that allows the adjustment of their values using reconstructed 3D images of reference structures as feedback.

Because it delivers the axial position of molecules from a single intensity measurement, SIMPLER is fully compatible with any 2D SMLM. SIMPLER–dSTORM delivers 3D images with sub-20-nm axial localization precision throughout the TIRF range of 250 nm, while for SIMPLER–DNA-PAINT, the axial localization precision is sub-10 nm. This level of axial resolution is only rivalled by

methods of high technical complexity, such as 4-Pi nanoscopy, MINFLUX or MIET. By contrast, SIMPLER requires no hardware modification whatsoever to a wide-field single-molecule fluorescence microscope and is highly robust; we validated its performance in both a custom-built and a commercial microscope. Furthermore, unlike other 3D fluorescence nanoscopy methods that require drift correction with nanometric precision to achieve nanometric resolution[17,23], SIMPLER only requires axial stability provided by any standard focus-lock system (~±100 nm over several hours). This is because the distance of the specimen to the interface (the dielectric substrate-sample interface) remains fixed independently of sample drift.

In summary, making quantitative use of a TIRF microscope in combination with the concepts of super-resolution microscopy, it is possible to locate single molecules with nanometric accuracy simply from a measurement of their emission intensity. Due to its robustness and practicality, SIMPLER can be directly applied by any lab counting with a conventional TIRF SMLM microscope, making 3D fluorescence nanoscopy readily available to numerous users and enabling a new wave of discoveries about the structure and pathways of sub-cellular structures and protein–protein interactions.

## Methods

**Simulation of single-molecule emission**. The angular emission pattern of single molecules was calculated using a Finite Difference Time Domain solver (CST Microwave Studio). The molecules were considered as a small (1 nm) dipole oscillating at the frequency of emission. The fraction of detected fluorescence was obtained by integrating the emission pattern over the solid angle of interest. The single-molecule images were obtained by focusing the fraction of the emission pattern collected by the objective. We provide sets of calculations for the most usual configurations in the Supporting Information, Supplementary Table 2.

**Super-resolution microscopy set-up 1**. The microscope used for TIRF SMLM data shown in Figs. 2 and 3 and Supplementary Figs. 3, 6, 7, 8, 9, 12, 13 was built around a commercial inverted microscope stand Olympus IX73 equipped with a high numerical-aperture oil-immersion objective lens (Olympus PlanApo 60x/NA 1.42). Excitation was carried out with a circularly polarized 642 nm 1.5-W laser (MPB Communications 2RU-VFL-P-1500-642). TIR illumination was achieved with a linear translation stage (Thorlabs MT1-Z8) used to control the lateral position of the focused excitation beam on the back focal plane of the objective. The angle of incidence was set to 69.5° (Supplementary Method 1 and Supplementary Fig. 4). A dichroic mirror (Semrock Di03-R 405/488/532/635-t1) and a band-pass filter (Chroma ET700/75m) were used to separate the fluorescence emission of the sample from the laser excitation. The emission light was expanded with a 2× telescope to optimize the pixel size of the EM-CCD camera (Andor iXon3 897 DU-897D-CS0-#BV) for single-molecule localization (133 nm in the focal plane). The camera and laser were controlled with Tormenta, a custom software developed in the laboratory and described in an earlier publication[61]. Typically, we acquired sequences of 50,000–100,000 frames at 4-Hz acquisition rate with a laser power density of ~2.5 kW/cm$^2$ for DNA-PAINT, and 50 Hz, and ~3 kW/cm$^2$ for dSTORM.

**Super-resolution microscopy set-up 2**. To demonstrate the ease of use of the technique, SIMPLER was directly applied to 2D SMLM images acquired with a commercial microscope Nikon N-STORM 5.0 located in the Nikon Imaging Centre at King's College London, UK and using the instrument internal calibration of the incidence angle (Fig. 4 and Supplementary Fig. 10). The microscope is equipped with a 100 × 1.49 NA oil-immersion TIRF objective, a perfect focus system for stable axial drift-free imaging, a gradient neutral-density filter to produce a near-flat-top intensity profile from the Gaussian-shaped beam input and an sCMOS camera (Orca Flash 4.0 V3, Hamamatsu). Samples were imaged under TIRF illumination with a 647-nm laser line that was coupled into the microscope objective using a quad-band set for TIRF (Chroma 89902-ET-405/488/561/647 nm). The final pixel size of the image was 160 nm in the focal plane. We acquired sequences of 50,000–100,000 frames at 10-Hz acquisition rate with a laser power density of ~2.5 kW/cm$^2$ for DNA-PAINT, and 50 Hz, and ~4 kW/cm$^2$ for dSTORM using an angle of incidence of 69º. Images were recorded with Hamamatsu's real-time hot-pixel correction enabled.

**TIRF angle calibration**. Incident angle of the excitation beam was determined by measuring the lateral displacement of the centre of fluorescence excitation upon axially translating the sample[42]. 1-micrometre Alexa Fluor 647 solution was illuminated with an incident angle $\theta_i > \theta_c$ and the sample was translated in z direction

from $z = 0$ to $z = 10\,\mu m$, in 0.4-µm steps (Prior ProScan III). As a consequence of the z-translation of the sample, the excitation spot was displaced in a lateral direction (y). The value of $\theta_i$ was obtained by fitting the dependence of lateral movement of the centre of the excitation beam on the z-translation with a linear regression, where $\Delta y = m\Delta z + c$ and $\arctan(m) = \theta_i$ (see calibration data and fit in Supplementary Fig. 4).

**Primary neuron culture and cell lines**. Mouse (CD1) hippocampal neurons were harvested from embryonic day 17 pups, following the general guidelines of the National Institute of Health (USA) and approval of the National Department of Animal Care and Health (SENASA, Argentina), and cultured in Neurobasal medium (Gibco) supplemented with 5-mM GlutaMAX-I (Gibco) and 2% B27 supplement (Gibco) at 37 °C and 5% CO$_2$. Mice were housed at 22–25 °C at a 40–60% humidity in a 12/12-h light/dark cycle. Neurons were seeded at a density of 125 cells/mm$^2$ on #1.5 thickness glass-bottomed chamber slides (Lab-Tek II, Thermo Fisher Scientific) and incubated for either 3 or 28 days, respectively. To increase cell attachment, glass slides were previously coated with 0.05-mg/mL poly-L-lysine (overnight at 37 °C) (Sigma Aldrich) and 1-µg/µL Laminin (3 h at 37 °C) (Sigma Aldrich).

Culture of COS-7, Human Fetal Foreskin Fibroblasts HFFF2 (ECACC 86031405) and HeLa Kyoto with endogenous Nup107 tagged with mEGFP (CLS Cell Lines Service GmbH) cell lines were grown in Dulbecco's modified Eagle's medium supplemented with 10% fetal bovine serum and 2-mM L-glutamine (Gibco) at 37 °C and 5% CO$_2$.

**Sample fixation and permeabilization**. Neurons, COS-7 and HFFF2 cells were fixed and permeabilized in PHEM buffer (60-mM PIPES, 25-mM HEPES, 5-mM EGTA, 1-mM MgCl$_2$, pH = 7.0), supplemented with 0.25% glutaraldehyde, 3.7% paraformaldehyde, 3.7% sucrose and 0.1% Triton X-100, for 20 min at room temperature. Auto-fluorescence was quenched by incubating the samples in 0.1-M glycine in PBS for 15 min followed by 3× washes with PBS. The fixed and quenched samples were blocked with 5% BSA in PBS containing 0.01% Triton X-100 for 1 h. HeLa Kyoto mEGFP-Nup107 cells were flash fixed and permeabilizing by sequentially using 2.4% paraformaldehyde in PBS for 30 s and 0.1% Triton X-100 for 3 min. After 3 × 5-min washes with PBS, cells were further fixed with 2.4% paraformaldehyde in PBS for 20 min, and auto-fluorescence was quenched by incubating the samples in 0.1-M glycine in PBS for 5 min followed by 3× washes with PBS.

**Immunostaining and imaging**. Spectrin in neurons (Fig. 2c and Supplementary Figs. 6, 8) was labelled with a mouse monoclonal primary antibody anti-β-Spectrin II (Clone 42/B-Spectrin II, BD Biosciences) for 1 h at room temperature using a 1:400 dilution in 5% BSA in PBS, followed by 3× washes with PBS. DNA-conjugated secondary antibody staining was performed by incubating the sample with a donkey anti-mouse secondary fragment antibody (Jackson ImmunoResearch, 715-007-003) at a 1:100 dilution in 5% BSA in PBS for 1 h at room temperature, followed by 3× washes with PBS. Microtubules in COS-7 cells (Fig. 3a and Supplementary Fig. 6, 13), HFFF2 cells (Supplementary Fig. 10) and in neurons (Supplementary Figs. 9 and 13) were treated with anti α-tubulin and anti β-tubulin primary antibodies for 1 h at room temperature using 1:400 dilutions in 5% BSA in PBS, followed by 3× washes with PBS (mouse monoclonal anti-α-Tubulin, clone TUB-A4A Sigma Aldrich; mouse monoclonal tyrosine anti-α-Tubulin, clone TUB-1A2 Sigma Aldrich; rabbit polyclonal anti-β-III-Tubulin, Abcam #ab 18207 for neurons; and rabbit polyclonal anti-β-II-Tubulin Abcam #ab 196 for COS-7 cells, a kind gift of Dr Jesus Avila, Centro de Biologia Molecular 'Severo Ochoa' CBMSO, Consejo Superior de Investigaciones, Cientificas, Universidad Autonoma de MadridUAM, C/Nicolas Cabrera, 1. Campus de Cantoblanco, 28049 Madrid, Spain[62]). In both cases, secondary staining was done by 1-h treatment at room temperature with a mix of donkey anti-mouse DNA-conjugated secondary fragment antibody (Jackson ImmunoResearch, 715-007-003) and donkey anti-rabbit DNA-conjugated secondary fragment antibody (Jackson ImmunoResearch, 711-007-003) at a 1:100 dilution in 5% BSA in PBS for DNA-PAINT imaging or Alexa-Fluor-647-conjugated goat anti-rabbit (Invitrogen, #A-21245) and Alexa-Fluor-647-conjugated goat anti-mouse (Invitrogen, #A-21235) secondary antibodies at a 1:300 dilution in 5% BSA in PBS for dSTORM imaging, followed by 3× washes with PBS.

Nup107 in HeLa Kyoto mEGFP-Nup107 cells were labelled with an Alexa Fluor 647 anti-GFP rat monoclonal primary antibody (Clone FM264G, BioLegend) for dSTORM imaging (Fig. 4a) by incubating overnight at 4 °C using a 1:200 dilution in 2% BSA in PBS, followed by 3× washes with PBS.

100-nanometres gold nanoparticles (BBI solutions) were added as fiducial markers for drift correction by incubating the sample for 5 min in a 1:2 solution of nanoparticles in PBS. After 3× washes with PBS, either PAINT buffer (Buffer B+: 5 mM Tris-HCl, 10 mM MgCl$_2$, 1 mM EDTA and 0.05 % Tween 20 at pH 8.0) containing fluorescently labelled DNA imager strands (Img1 and Img2, biomers. net GmbH, for imaging anti-mouse and anti-rabbit DNA-conjugated secondary antibodies, respectively; see Supplementary Table 3) or STORM imaging buffer containing 50-mM TRIS pH 8, 10-mM NaCl, 10% w/v D-glucose, 10-mM mercaptoethylamine, 1-mg/mL glucose oxidase and 40-µg/mL catalase were added to the immunolabeled samples.

Antibody conjugation to DNA-PAINT docking sites (Dock1 and Dock2, biomers. net GmbH, for the donkey anti-mouse and the donkey anti-rabbit conjugates, respectively; see Supplementary Table 3) was performed using maleimide-PEG2-succinimidyl ester coupling reaction according to a published protocol[63] as described in Supplementary Method 1. Imager strands concentrations were 20-nM Img1 for spectrin imaging (Fig. 2c and Supplementary Figs. 6, 8), and 80-pM Img1 + 80-pM Img2 for α-tubulin + β-tubulin imaging (Fig. 3a and Supplementary Figs. 6, 9, 10). Samples were then used immediately for DNA-PAINT imaging.

**Data acquisition, analysis and 3D image rendering.** Lateral $(x, y)$ molecular coordinates and photon counts $(\widehat{N})$ were obtained using the Localize module of Picasso software[63] and enabling the symmetric PSF fitting method. Drift correction was carried out with a combination of redundant cross-correlation and fiducial markers approach using the Render module of Picasso. Photon counts were corrected by using the illumination profile of the beam, which was either measured by imaging a 1-μM Alexa Fluor 647 solution with the same incident angle as in the biological experiments or by approximating it with the corresponding background parameter obtained from the MLE analysis.

Next, localizations were filtered to discard the frames corresponding to the switching (ON or OFF) of the fluorophores during the frame acquisition, whose photon count would be lower and lead to falsely high $z$ coordinates. To ensure the molecule emitted during the whole exposure time, localizations were kept as valid only in the case that other localizations, reasonably attributed to the same fluorophore (within 3–5 nm i.e. 20 nm), were detected in the previous and subsequent frames (Fig. 2a). Therefore, a localization $(x_n, y_n)$ found in the frame $n$ was kept as valid only if there was another localization in frame $n-1$ located at $\left[ (x_n - x_{n-1})^2 + (y_n - y_{ln-1})^2 \right] < (20\,\text{nm})^2$ and another localization in frame $n+1$ located at $\left[ (x_n - x_{n+1})^2 + (y_n - y_{n+1})^2 \right] < (20\,\text{nm})^2$. Molecules detected for less than three frames were thus ignored.

For each image, a photon count was assigned to $z = 0$ $(N_0)$. This value was obtained directly by measuring samples of Fab fragments or antibodies adsorbed to the coverslip. Experimental measures of the $z$-localization precision $(\sigma_z)$ were determined from the variance of $z$-localizations of the same molecule. Values of $\sigma_z$ were registered for multiple molecules located at different $z$-positions as described in the main text (Figs. 3e and 4d).

The software used to analyse SMLM data and obtain the axial positions of single molecules through SIMPLER was custom-built in MATLAB.

$z$-colour-coded image rendering was done using the ImageJ plug-in ThunderStorm[64], importing the list of all $(x, y, z)$ coordinates without merging localizations re-appearing in subsequent frames. A Gaussian filter with a size proportional (factor 0.5) to the median localization precision in the 0–250-nm axial range $(\sigma = 2\,\text{nm}$, for DNA-PAINT data and $\sigma = 6\,\text{nm}$ for dSTORM data) was used for all three dimensions. A lenient density filter was applied to $xy$ images, to discard localizations with <100 neighbours in a 67-nm radius, to enhance contrast by suppressing some of the non-specific localizations of the background.

Image-based resolution was computed from rendered images using the Fiji plug-in of image decorrelation analysis[58]. For this, images were rendered by adding the contribution of each $yz$ or $xz$ localization as a 2D Gaussian function with a global standard deviation equal to the median axial localization uncertainty: 4 nm for DNA-PAINT data (Fig. 3) and 12 nm for dSTORM data (Fig. 4).

**DNA-antibody coupling reaction.** DNA labelling of a fragment secondary antibody (donkey anti-mouse IgM, 715-007-003 or donkey anti-rabbit IgM, 711-007-003, Jackson ImmunoResearch) was performed using the maleimide-PEG2-succinimidyl ester coupling reaction[63]. In order to reduce the thiolated DNA for the maleimide reaction, 15 μL of 1-mM thiol-DNA (Dock1 and Dock2 biomers.net GmbH, for the donkey anti-mouse and the donkey anti-rabbit conjugates, respectively; see Supplementary Table 3) were incubated with 35 μL of 250-mM DDT (Thermo Fisher Scientific) freshly prepared solution (1.5-mM EDTA, 0.5x PBS, pH 7.2) on a shaker, in the dark, for 2 h at room temperature. 30 minutes after the reduction of the thiol-DNA started, 30 μL of 26-μM fragment antibody was incubated with 0.7 μL of 23.5-mM Maleimide-PEG2-succinimidyl ester (Sigma Aldrich) solution on a shaker, in the dark, for 90 min at 4 °C. Prior DNA-antibody conjugation, both sets of reactions were purified using an illustra MicroSpin G-25 column (GE Healthcare) to remove excess of DDT and a Zeba desalting column (Thermo Fisher Scientific) to remove excess of cross-linker. Next, both flow-through of the columns were mixed and incubated on a shaker, in the dark, overnight at 4 °C. The next day, DNA excess was removed by Amicon spin filtration (30 kDa). Antibody-DNA concentration was measured with the NanoDrop spectrophotometer and adjusted to 14 μM with PBS. DNA-labelled antibodies were stored for a maximum of 6 months at 4 °C.

**Reporting summary.** Further information on research design is available in the Nature Research Reporting Summary linked to this article.

## Data availability
Several localization datasets are included as example data to run the SIMPLER Supplementary Software, and are available on Github: https://github.com/ cibion-conicet/SIMPLER. The authors also uploaded source data and images to the public repository FigShare (https://doi.org/10.6084/m9.figshare.13326599). Other data are available from the corresponding authors upon reasonable request.

## Code availability
The MATLAB software code to apply SIMPLER is freely available as Supplementary Software and at https://github.com/cibion-conicet/SIMPLER. The DOI for the software (Version 1.0.0) is https://doi.org/10.5281/zenodo.4300989[65]. An executable program is also available from the corresponding author upon reasonable request.

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

## Acknowledgements

This work has been supported by: CONICET, ANPCYT projects PICT2013-0792 and PICT-2014-0739, the Royal Society project IEC\R2\181018, the BBSRC grant BB/R007365/1, FOCEM (Fondo para la Convergencia Estructural del Mercosur) grant COF 03/11, and Swiss National Science Foundation through the National Center of Competence in Research Bio-Inspired Materials. S.S. acknowledges financial support from the Human Frontier Science Program Organization and the Royal Society through a HFSP (LT001463/2017-C) and Dorothy Hodgkin (DHF\R1\191019) fellowship, respectively. F.D.S. thanks the support of the Max-Planck Society and the Alexander von Humboldt Foundation. D.R. acknowledges the support of the Max-Planck Society and the Volkswagen Stiftung.

## Author contributions

S.S. and F.D.S. conceived the approach, discussed results and wrote the manuscript with input from all authors. A.M.S., B.S., S.S. and F.D.S. developed the analysis method. A.M. S., B.S. and S.S. acquired and analysed the data. D.J.W. helped with the data analysis. D. M.O. discussed results and commented on the manuscript. A.M.S. and S.S. produced the custom-made DNA-PAINT antibodies. J.L., N.U., D.R. and A.C. contributed the primary neuron cultures and COS-7 cell lines, and carried out the spectrin and microtubules immunostaining. F.D.S., M.P.-P. and G.A. simulated the emission pattern of single molecules. A.M.S. wrote the analysis scripts and designed the GUI. S.S. and D.J.W. carried out the microtubules immunostaining in HFFF2 cells and the nucleoporin Nup107 in HeLa Kyoto cells.

## Competing interests

The authors declare no competing interests.
