## [Peer Review File · Nature Communications]

REVIEWER COMMENTS

Reviewer #1 (Remarks to the Author):

In their manuscript about SIMPLER Szalai et al present a very interesting SMLM method that promises high localisation precision. I am favourably impressed by the approach but would be keen to see a number of clarifications and further testing/validation. In addition, some ways to make the essence of the approach clearer and more readable would help. Specifically, I would like to raise the following points:

- throughout the paper the term "resolution" is used where it most of the time actually refers to the achievable localisation precision; as super-resolution imaging has matured the concepts of single-molecule localisation precision and imaging resolution have been recognised to be clearly distinct; for example, resolution can be estimated with such methods as FRC/FSC (Fourier Ring/Shell Correlation) and is almost always significantly lower than the localisation precision. This needs to be carefully rectified throughout the manuscript and the distinction maintained and emphasised.

- in connection with this point, the authors should attempt to measure the FSC of some of their data sets to provide actual resolution estimates

- while the use of biological samples to illustrate the utility of SIMPLER is acknowledged by this reviewer, it would be desirable to also investigate SIMPLER with a specifically designed 3D test sample, such as the 3D origami samples that are now commercially available; this would allow directly measuring axial (and lateral) localisation precision at single locations where origami binding sites are positioned in 3D; in addition, these rods in 3D have a variable angle so that different depth are probed with the ensemble of 3D origami; this would allow a more detailed experimental characterisation of SIMPLER, in addition to the biological samples.

"Furthermore, unlike other 3D fluorescence nanoscopy methods, the level of resolution achieved by SIMPLER does not depend on nanometric axial drift corrections. This is because the measurement reference (the dielectric substrate-sample interface) is part of the sample." This is not really clear to this reviewer. How do you know where the interface is if, for example, you have no stain at that level which could easily occur with some biological targets; please clarify and point out the experimental procedure that was used to locate the interface.

- The sequence of calibration measurements/parameter characterisations to implement SIMPLER are described in the text but it is difficult for the reader to get a clear overview in one place that she/he could use like a checklist or protocol. I would suggest that such a protocol or checklist is added as a supplementary item as it would make it easier for readers to adopt SIMPLER

- similarly, and in conjunction with such a protocol it would be useful to list the key parameters in a table and to list how critical their exact determination is for achieving high localisation precision; such a table would probably be useful in the main text as an easy go-to place to get the bigger picture

- the procedure to only use events that occur in a burst and discard first and last events because they may not last throughout the whole frame seems reasonable at first sight; however, it has been observed that, even in DNA-PAINT with dyes chosen for photo-stability (rather than switching), fast "flickering" can occur throughout a burst, i.e. dye blinking that can be faster than the frame duration. How can such an effect be excluded and, if not, how can one be sure that it has no major effect on the localisation precision. (In this context experiments with 3D origami could prove valuable as suggested above).

Minor:

- "invariability of the shape of the single molecule images"; formulated in more conventional

terms, I imagine you are referring to the effective lateral point-spread function being close to constant over the thickness of the usable depth (~250 nm). This seems expected, the additional impact of the illumination function is a scaling that due to the stratified nature of the illumination field is simply an intensity scaling. It might be useful to discuss in these or similar terms.

"in good agreement with what is expected for an immunolabeled microtubule (primary antibodies and Fab fragments from secondary antibodies)." Please provide suitable references.

On pg 14: "As an example, Figure 2d shows the normalized profiles of average signals..." this seems to refer to Fig. 2e.

Reviewer #2 (Remarks to the Author):

In this manuscript Szalai et al report on a methodology based on total internal reflection microscopy (TIRF) to obtain sub-10nm axial nanometric resolution over a depth of around 250nm from the glass interface, in combination with single molecule localization DNA-PAINT or dSTORM. The method is in fact extremely simple and already inherent to current SMLM set-ups, thus straightforward to implement. It is very well known in the community that by proper calibration of the evanescent field in the z-direction one could determine the axial position of an object with higher resolution than that fixed by diffraction. Yet, the authors do a proper job by making a quantitative study on the dependence of the evanescent field in the axial direction, as a function of several parameters and combine it with the concepts of SML to obtain nearly isotropic nanometre resolution in 3D. Of particular relevance, the authors consider in their model the angular emission of individual molecules to account for variations in the detected fluorescence signal. The manuscript is well written, the data are solid and importantly in my opinion, the simplicity of this method makes it useful to the community.

There are pro's and con's of the method: The major advantages rely on the simplicity, since it does not require a different set-up, neither adaptation of a current SMLM, i.e., basically all the SMLM methods work under TIRF illumination. The resolution is superior that most commonly used methods (astigmatism or DONALD) and comparable to that of iPALM or iSTORM, but orders of magnitude simpler to implement. On the other hand, it bears an inherent limitation: the fact that the method is essentially restricted to axial distances below than 250nm (the penetration depth of the evanescent field). Considering recent efforts in the field to achieve comparable axial nm resolution deeper into the sample and/or to extend the axial working range to several microns, the excitement for this work by the community might be modest.

I have a few remarks that would need to be addressed in a revised version:

1) In the first part of the paper, the authors rightly highlight the importance of including the angular emission pattern of individual dipoles in the calculation of the collected fluorescence intensity. Indeed in Figure 1c,d, they estimate the emission pattern of molecules according to their dipole orientation with respect to the glass-water interface. However, the equations that follow later and used in the ms to obtain the parameters needed for determining axial positions, only consider an average dipole emission, assuming that fluorophores will rotate freely during the time scale of the experiments. I am not fully understand why they make this assumption. Most super-resolution experiments are performed on fixed samples, where most probably the fluorophores are not freely rotating and therefore having different fixed dipole emissions. Thus, either the authors experimentally demonstrate free rotation of fluorophores on fixed samples so that their equations are valid, or, would need to adapt their model to include the influence of dipole emission. In the latter case, I guess they would need to have prior knowledge of the dipole emission of each detected molecule, which would add complexity to the set-up.

2) The authors argue that the method is easy to implement and provide a look up table for users. Nevertheless, the accuracy of the method will ultimately depend on the quality of the set-up and calibration might not always be easier & comparable from one set-up to another. I suggest that for completeness the authors add a paragraph on the discussion regarding a word of caution to users

and how critical the method is to inaccuracies of the set-up and/or the prior parameters to be determined.

3) Figure 1g: what are the different lines (and colours) associated to SIMPLER? Very difficult to see & distinguish the different colours. The text associated to this figure (page 9) is also very confusing: comparison between dF & No for different axial lengths... how do these numbers relate to the figure?

4) Page 11: Explanation of Figure 2c in the text is also confusing: where does the 10nm average accumulated distortions within 0-250 axial range comes from?

5) Page 11: last sentence: what do the authors mean by "almost fixed xy position".????

6) Figure 3d: The caption is not clear enough.

7) Page 17: What do they mean by the "complete working range"?, Are the authors referring to 0-250nm axial range?

Reviewer #3 (Remarks to the Author):

The manuscript by Szalai, et al. describe a new approach called SIMPLER used to estimate the axial position of single molecules in dSTORM when performed using TIRF imaging. SIMPLER relies on the intensity of the single molecules showing a dependence on the z-position in the evanescent wave. Using the intensity of molecules at the coverglass as a reference, the signals can be converted into axial positions and plotted in three dimensions. The reported axial resolutions, which can be achieved on relatively simple or at least commercially available microscopes using existing image analysis software, rival or best some of the most precise single molecules methods based on more advanced instrumentation. The resulting data and final images, such as the cross-sections of microtubules, are compelling and suggest the authors have developed an approach which can be adopted by almost any scientist with access to a TIRF microscope. However, I am unclear about how the uncertainties in the axial positions of the molecules are determined and displayed for several examples. I am also unclear why the uncertainty for some parameters are not factored into the final axial position uncertainty. Given the potential power and widespread use of this approach and that the precision of localization is arguably one of the most critical results provided by SIMPLER, adequate answers or well-defined arguments to the points listed below will hopefully clarify some of the key features of this method.

- The photons per molecule per frame in figure 2b shows a much larger mean and a much smaller distribution than other investigators have typically measured for single molecules of Alexa647 and ATTO655. Is this simply due to the selection of molecules which last 3 or more frames? It would be helpful if the authors showed examples of unfiltered histograms. This might also help readers appreciate how many molecules are excluded from the final images and the authors should provide this information for each figure. Moreover, plotting a few of the images with all of the molecules would provide a helpful comparison.
- How are the molecules plotted? From the methods section, I understand that a 2nm Gaussian blur was applied to all directions. This implies a precision of 2nm or less, but this is not the axial resolution described throughout the full TIRF range in the text. Moreover, results presented in figure 3e indicate that plotting with a Gaussian blur should be performed with a much larger sigma.
- Can the authors also clarify how the lateral and axial positions for molecules lasting more than 3 frames are determined. Are the positions determined from the average of the positions in each frame? From the average of the photons per frame for axial? Summing the photons across frames for lateral?
- In the comparison of filtered and unfiltered data in supplementary figure 5, the number of molecules should remain constant while the molecule positions in the unfiltered should simply be shifted to a higher axial position due to a decrease in their average number of photons per frame. However, the unfiltered data images seem to show a lot more molecules than the filtered data images. In both examples, it is unclear from where in the filtered image the molecules are being axially translated when plotted in the unfiltered images.
- Clarification on the estimation of the variances used will also be helpful. Most of the points below concern this critical part of the method.

- Can the authors please define the variances estimated and used in the error determination for each of the figures in which example images are plotted?
- I do not understand the results in Figure 1e which is supported by Supplementary figure 1a. From my reading of the text, the exact solution represents the product of the red line CF in figure 1d and the evanescent wave approximation (the blue line in figure 1e). When normalized at $z=0$, shouldn't the exact solution always display a steeper decline and a lower amplitude than the evanescent wave approximation at all values $z>0$?
- I am unconvinced that the uncertainty for the N_0 value is of minor importance as indicated in the text. Could the authors elaborate and help clarify their argument for this conclusion? I understand from the Figure 2c and supplementary figure 2 that choosing a value of N_0 that does not correspond to $z=0$ will lead to an offset in the position of the other molecules. But shouldn't this be plotted as an uncertainty in the position of the molecules rather than an offset? The histogram in figure 2b should provide σ_{N_0} , correct?
- The authors indicate that the lower bound for the uncertainty might be estimated from the square root of the number photons. Figure 2b should offer an adequate test for this approximation. How well does the data in figure 2b approximate this theoretical lower bound? If the approximation is insufficient, the authors should use the uncertainty of the reference molecules located at $z=0$ as the practical lower bound.
- From analyzing single molecule traces, the authors conclude that the photon uncertainty for molecules $z>0$ can be estimated from $5\sqrt{N}$. Does this hold for N_0 as well?
- Some of the simulations (such as Figure 1g) indicate the axial uncertainty approaches zero as the axial position approaches $z=0$. Given that the authors find in figure 3 that $5\sqrt{N}$ seems to offer a good approximation for axial uncertainty, it is unclear how it can approach zero even at $z=0$. Can the authors clarify or explain this?
- The authors simulate the expected uncertainty for many of the components in equation 3 used to determine the axial positions (Fig. 1g, Fig. 2c, Supp. Fig 1, Supp. Fig. 2). Generally, the components of interest in each figure introduce $<10\text{nm}$ uncertainty over the axial range, but it is unclear what uncertainties are assumed for the other components when making these simulations. It will be helpful if the authors include this information in the legend for each figure?
- The parameters d_f and a_f are derived from a fit of $I(z) \times CF(z)$, both of which are calculated based on instrument and sample parameters. The use of just these calculated values is worrisome. It would be more compelling if the authors offered experimental determinations for the values using their imaging systems to compare with the simulated values.
- It is unclear how the uncertainty for the TIRF depth, σ_{d_f} , and the uncertainty for the non-evanescent component, σ_{a_f} , are estimated in each experiment. Given they are required to calculate the axial position uncertainty, good estimates for these values are likely critical. The authors have included simulations for these which indicate they have little bearing on the axial position uncertainty. However, it is unclear if those simulated uncertainties reflect uncertainties in an experimental setting. Could the authors please elaborate on the estimation of these uncertainties?

Minor points

- Page 22; sentence 2. Typo.
- For one of the super-resolution microscopes, a sCMOS camera was used in the single molecule localization experiments, but it was unclear from the methods if the corrections for the pixel dependent noise common in these cameras was applied. Given the noise characteristics for these cameras, this correction is usually required for single molecule imaging. If they authors performed these corrections, please indicate this in the methods. If not, the authors may wish to revisit data from images produced by this instrument.

Response to reviewers

NCOMMS-20-13404-T

We thank the three Reviewers for the sharp reading and positive appraisal of our paper. Their constructive comments have helped us to improve the clarity of our manuscript.

Below, we give a point by point response to the comments, and we note when modifications were done to the paper.

Reviewer's comments in *italic*

Reviewer #1 (Remarks to the Author):

In their manuscript about SIMPLER Szalai et al present a very interesting SMLM method that promises high localisation precision. I am favourably impressed by the approach but would be keen to see a number of clarifications and further testing/validation. In addition, some ways to make the essence of the approach clearer and more readable would help. Specifically, I would like to raise the following points:

Comment (I-1): *1- throughout the paper the the term “resolution” is used where it most of the time actually refers to the achievable localisation precision; as super-resolution imaging has matured the concepts of single-molecule localisation precision and imaging resolution have been recognised to be clearly distinct; for example, resolution can be estimated with such methods as FRC/FSC (Fourier Ring/Shell Correlation) and is almost always significantly lower than the localisation precision. This needs to be carefully rectified throughout the manuscript and the distinction maintained and emphasised.*

Response (I-1): Reviewer #1 is right to point out that we have used the term ‘resolution’ in contexts where the term ‘localization precision’ should have been employed instead. We have rectified this throughout the manuscript.

Comment (I-2): *2- in connection with this point, the authors should attempt to measure the FSC of some of their data sets to provide actual resolution estimates*

Response (I-2): Following the suggestion of Reviewer #1, we calculated the resolution of the y/z and x/y cross-sections of microtubule data presented in Figure 3 and nuclear pore complexes presented in Figure 4 (Figure R1, new Supplementary Figure 11) using a recently reported parameter-free image resolution estimation method based on decorrelation analysis (Nature Methods,16, 918–924, 2019). We opted to use this algorithm for assessing the resolution of the individual super-resolved images as it is model-free and does not require any user-defined parameter. Furthermore, in contrast to FRC, this method is independent of the probability of multi-blinking. We note that the effects of “multiblinking”, either due to the photoswitching kinetic in dSTORM or due to multiple binding and unbinding in DNA-PAINT, can severely

impact the FRC/FRS resolution estimate by introducing spurious correlations, which can only be mitigated if an accurate estimation of the multiple blinking statistics is available (Nature Methods, 10, 557-562, 2013).

We note that resolution estimates in SMLM images depend on the parameters used to render the super-resolved images from the localization data. Therefore, for resolution estimation, we have rendered images using all valid localizations, each one plotted as a Gaussian with standard deviation equal to the median axial localization uncertainty: 4 nm for DNA-PAINT data (Figure 3) and 12 nm for dSTORM data (Figure 4). The resolution estimate was 11 nm for DNA-PAINT images and 34 nm for dSTORM images.

Figure R1. Image decorrelation analysis. Decorrelation functions computed for the image-based resolution estimation for (a) DNA-PAINT images presented in Figure 3 and (b) dSTORM images presented in Figure 4. Green, decorrelation function without any high-pass filtering; grey, decorrelation functions with high-pass filtering; cyan lines, decorrelation functions with refined mask radius and high-pass filtering range; blue triangles, local maxima. Vertical line, cut-off frequency. C.c., cross-correlation.

In addition, we have included the following sentences in the main manuscript text referring to the estimation of image-based resolution for DNA-PAINT and dSTORM images respectively:

“The average axial localization precision is well below 10 nm throughout the complete z range. This observation was also supported by image-based decorrelation analysis (Nature Methods,16, 918–924, 2019) of microtubules cross-section presented in Figure 3a, resulting in an estimated average spatial resolution of ~ 11 nm (Supplementary Figure 11a).”

“The experimentally determined axial localization precision (161 single-molecule traces) was found to be below 20 nm throughout the 0 - 250 nm axial range (Figure 4d), while the spatial resolution determined through image decorrelation analysis (Nature Methods,16, 918–924, 2019) was ~ 34 nm (Supplementary Figure 11b).”

And in the Methods section:

“Image-based resolution was computed from render images using the Fiji plugin of image decorrelation analysis (Nature Methods,16, 918–924, 2019). Images were rendered by adding the contribution of each yz

or xz localization as a 2D Gaussian function with a global standard deviation equal to the median axial localization uncertainty: 4 nm for DNA-PAINT data (Figure 3) and 12 nm for dSTORM data (Figure 4).”

Comment (I-3): 3- *while the use of biological samples to illustrate the utility of SIMPLER is acknowledged by this reviewer, it would be desirable to also investigate SIMPLER with a specifically designed 3D test sample, such as the 3D origami samples that are now commercially available; this would allow directly measuring axial (and lateral) localisation precision at single locations where origami binding sites are positioned in 3D; in addition, these rods in 3D have a variable angle so that different depth are probed with the ensemble of 3D origami; this would allow a more detailed experimental characterisation of SIMPLER, in addition to the biological samples.*

Response (1-3): - We thank Reviewer #1 for this suggestion, which is good. In fact, we have been using DNA-origami every time we found them suitable for our projects for almost a decade, both to construct optical nano-antennas and to test super-resolution imaging (ACS Nano 6 (2012) 3189–3195, Nano Letters 14 (2014) 2831–2836, Science 355 (2017) 606–612, Nature Communications 8 (2017) 13966, Nano Letters 19 (2019) 6629–6634). For this project too. We have considered using them from the beginning. However, already from preliminary experiments, we found that using regular biological structures of well-known geometry was more efficient. In this respect, our work is in line with the recent proposal by the group of Jonas Ries and collaborators of using such structures as standards for super-resolution microscopy (Nature Methods 16 (2019) 1045–1053).

The microtubule cross-sections or the nuclear pore complex present well-defined geometries that are more rigid and stable than DNA-origami. Also, for our experiments, we can easily find them (and check their correct visualization) at different z coordinates throughout the TIRF range, which is highly challenging for DNA-origami structures.

Comment (I-4): 4- *“Furthermore, unlike other 3D fluorescence nanoscopy methods, the level of resolution achieved by SIMPLER does not depend on nanometric axial drift corrections. This is because the measurement reference (the dielectric substrate-sample interface) is part of the sample.” This is not really clear to this reviewer. How do you know where the interface is if, for example, you have no stain at that level which could easily occur with some biological targets; please clarify and point out the experimental procedure that was used to locate the interface.*

Response (1-4): We were not sufficiently clear on this statement. We apologize. The passage mentioned by Reviewer #1 was not meant to imply that one needs to know the exact coordinate of the interface. On the contrary! What we mean is that even if the whole sample drifts with respect to the objective, the distance of the specimen to the interface remains unchanged, and therefore SIMPLER works just as fine. A conventional focus-lock is sufficient to keep the sample in place with enough precision for SIMPLER to work. This is in contrast to, for example, MINIFLUX or 4-Pi nanoscopy, which require drift corrections with nanometric precision to achieve nanometric resolution. We have rewritten the passage to make this clearer:

“Furthermore, unlike other 3D fluorescence nanoscopy methods that require drift correction with nanometric precision to achieve nanometric resolution (Cell, 166, 1028-1040, 2016; Nature Methods, 17, 217-224, 2020), SIMPLER only requires axial stability provided by any standard focus-lock system ($\sim \pm 100$ nm over several hours). This is because the distance of the specimen to the interface (the dielectric substrate-sample interface) remains fixed independently of sample drift.”

***Comment (1-5):** 5- The sequence of calibration measurements/parameter characterisations to implement SIMPLER are described in the text but it is difficult for the reader to get a clear overview in one place that she/he could use like a checklist or protocol. I would suggest that such a protocol or checklist is added as a supplementary item as it would make it easier for readers to adopt SIMPLER*

Response (1-5): This is an excellent suggestion by Reviewer #1, and in line with Comment 2-2 by Reviewer #2, which has led us to make a work-flow diagram summarising the information presented in the manuscript.

In the new Supplementary Figure S2 (Figure R2) we show the work-flow protocol of SIMPLER. In the diagram, we have differentiated the steps that are common to any SMLM method, from the additional calibration or analysis steps required for SIMPLER. Also, we have re-written the first paragraphs of the “Experimental implementation of SIMPLER” in the main manuscript. We believe this new diagram and the corresponding text modifications make it much clearer for the reader what it takes to adopt SIMPLER in their labs.

In addition, to ease the implementation of SIMPLER to the wide-imaging community, we have also expanded Supplementary Software 1 with an intuitive graphical user interface, so that users can directly compute z from their list of single-molecule localization data. The provided software will allow new users to easily apply and test SIMPLER. We also provide example data. Among other features, the software allows the determination of N_0 from a calibration sample SMLM data, it can correct photon counts for uneven illumination, and perform the frame filtering step.

Figure R2. SIMPLER protocol workflow. Supplementary Figure 2. SIMPLER protocol workflow. Starting with sample preparation, the user should prepare an additional sample to calibrate N_0 for the same imaging conditions as for the biological experiments. One option to prepare such a sample is to simply deposit the same fluorescent label used for biological imaging on a coverslip. Next, data acquisition is performed as in any typical 2D SMLM method

with the caveat of adjusting the acquisition/experimental conditions (power, frame rate, dSTORM switching buffer, DNA imager/docking sequence pair) so that the average single-molecule emission event last at least three camera frames. During image analysis, single-molecule fluorescence events are localized, drift-correction procedures are applied and photon counts can be corrected for uneven illumination. Then, special emphasis is given to filtering the localization list to exclude the first and last frames of each single-molecule emission event. To convert the number of emitted photons to z -positions, three parameters are needed: N_0 , d_F and α_F . N_0 is obtained by data analysis of the calibration sample. d_F and α_F are obtained from a fit to $F(z) = I(z) \times CF_{avg}(z)$, which is defined by the microscope set-up and sample/imaging conditions. The only experimental requirement to estimate these two parameters is to determine the angle of incidence of the excitation light θ_i . While some commercial set-ups already provide this value, a simple option to do this is to use the displacement method as described in Supplementary Method 1. SIMPLER is quite robust against mistaken values of α (see Supplementary Figures 1, 5, and 6). To simplify the adoption of SIMPLER, we provide Supplementary Software 1 and example data. The software also permits the adjustment of the calibration parameters using images of reference structures, such as microtubule cross-sections or nuclear pore complexes.

Comment (1-6):- *similarly, and in conjunction with such a protocol it would be useful to list the key parameters in a table and to list how critical their exact determination is for achieving high localisation precision; such a table would probably be useful in the main text as an easy go-to place to get the bigger picture*

Response (1-6): We acknowledge Reviewer #1 for pointing this out as it allows us to provide an easy-to-read table that summarises the influence of each of the parameters on the axial estimation. This issue was also raised by Reviewer #2 in Comment 2-4.

We have included the following table (Table R1 / Supplementary Table 2) in the revised Supplementary Information, along with new Supplementary Figures 1, 5 and 6 (Figures R3, R4, and R5), describing and computing examples of the axial mislocalization (off-set) and distortions introduced when using wrong calibration parameters. These figures show that the mislocalization and distortions are small for sensible ranges of incorrect values of the calibration parameters.

Parameter	Description	Main Effect when using an incorrect value
N_0	Emitted photons per frame by a fluorophore at $z = 0$	Axial off-set. Axial distortions far from the surface ($z > 150$ nm); flattening if N_0 is underestimated, or elongations if N_0 is overestimated.
θ_i	Angle of light incidence	Axial distortions; elongations if θ_i is underestimated, or compressions if θ_i is overestimated.
$1 - \alpha$	Scattering contribution factor	Axial distortions far from the surface ($z > 150$ nm); flattening if α is overestimated, or elongations if α is underestimated.

Table R1 (Supplementary Table 2). Effect of calibration parameters for axial determination via SIMPLER. Quantitative examples are shown in Supplementary Figures 1, 5, and 6.

Figure R3 (Supplementary Figure 1). Quantification of the axial mislocalization (Δz) when using the exponential approximation, or incorrect calibration parameters θ_i , α and N_0 (a) Exact solution and exponential fit of $F(z)$ for the experimental conditions of our experiments ($\theta_i = 69.5^\circ$, $\alpha = 0.90$). Inset: Δz between the curves for the range of z from 71 to 78 nm. Bottom: Δz between the exact solution and the exponential approximation as a function of z . For $z < 200$ nm, $\Delta z < 6$ nm. In the range of 0-150 nm, $\Delta z < 1$ nm. (b-d) Exponential $F(z)$ (top) and Δz (bottom) as a function of z , for ranges of θ_i , α and N_0 around the correct experimental values. (b) $\theta_i = \{68^\circ, 68.5^\circ, 69^\circ, 69.5^\circ, 70^\circ, 70.5^\circ, 71^\circ\}$. An incorrect θ_i by $\pm 1.0^\circ$ leads to $\Delta z < 7$ nm for $z < 150$ nm, and $\Delta z < 13$ nm for $z < 250$ nm. (c) $\alpha = \{0.88, 0.89, 0.90, 0.91, 0.92\}$; i.e. a range corresponding to a non-evanescent illumination component ($1 - \alpha$) of 12% and 8% of the total power at $z = 0$. A 10% incorrect α generates $\Delta z < 10$ nm in all the range from 0-250 nm. (d) $N_0 = \{0.8, 0.85, 0.90, 0.95, 1.0, 1.05, 1.10, 1.15, 1.20\}N_0$. An overestimation of N_0 by 10%, leads to $\Delta z = 8.5$ nm at $z = 0$ and $\Delta z = 18.1$ nm at $z = 250$ nm.

Figure R4 (Supplementary Figure 5). Example percentual axial distortions introduced when using incorrect calibration parameters. (a-c) Percentual axial distortion of the SIMPLER image of a microtubule (41 nm diameter),

centred at different axial positions, when incorrect values of N_0 (left), α (centre) and θ_i (right) are used. (d) Analogous calculations for a structure with 100 nm axial length. The values for each of the parameters are: (a) and (d) $N_0 = 50,000$ photons/frame; $\alpha = 0.9$ and $\theta_i = 69.5^\circ$; (b) $N_0 = 10,000$ photons/frame; $\alpha = 0.9$ and $\theta_i = 69.5^\circ$; (c) $N_0 = 50,000$ photons/frame; $\alpha = 0.9$ and $\theta_i = 67^\circ$.

Figure R5 (Supplementary Figure 6). Examples of SIMPLER reconstructions using different computation methods and varying α , θ_i , and N_0 . Side views (*i.e.* z - y projections) of a spectrin ring (bottom) and a microtubule (top) obtained with different z -computation approaches. In the first images (left), z was computed numerically using the exact solution and $\alpha = 0.90$, $N_0 = 50,000$, and $\theta_i = 69.5^\circ$ as calibration input parameters. Right next to them, the results obtained with the exponential approach and the same input parameters are shown. Next, 6 different computations of the same data are shown for each structure, varying only one calibration input parameter per image ($N_0' = \{0.8, 1.2\} N_0$ for the first two examples; $\theta_i' = \{68.5^\circ, 70.5^\circ\}$ for the third and fourth examples; and $\alpha' = \{0.88, 0.92\}$ for the last two cases). No significant axial distortions are observed over this range of parameters. Scale bars represent 50 nm (microtubules) and 100 nm (spectrin rings).

In the revised version of our paper, we are referencing to these figures and table when we finish describing how to obtain the calibration parameters:

“Overall, the robustness of SIMPLER against incorrect values of the calibration parameters θ_i , α and N_0 , is described in Supplementary Table 2 and Supplementary Figures 1, 5 and 6.”

Comment (I-7): - *the procedure to only use events that occur in a burst and discard first and last events because they may not last throughout the whole frame seems reasonable at first sight; however, it has been observed that, even in DNA-PAINT with dyes chosen for photo-stability (rather than switching), fast “flickering” can occur throughout a burst, i.e, dye blinking that can be faster than the frame duration. How can such an effect be excluded and, if not, how can one be sure that it has no major effect on the localisation precision. (In this context experiments with 3D origami could prove valuable as suggested above).*

Response (I-7): Referee #1 is right that an additional fast blinking could, in principle, lead to a higher variance of single-molecule intensity measurements (higher $\sigma_{\hat{N}}$). We note, however, that this could only occur for a quite limited range of blinking rates, namely if the faster blinking takes place with rates comparable to the camera frame rate (if it is faster, the end effect is just a lower average emission intensity. If it is slower is just as useful for SMLM).

The effect of $\sigma_{\hat{N}}$ on the localization precision σ_z is given by equation (3).

As a matter of fact, we determined $\sigma_{\hat{N}}$ experimentally and found it to be 5x larger than the ideal value of $\sqrt{\hat{N}}$. We thank Reviewer #1 for pointing out that micro-blinking could be, at least partly, responsible for this.

We have modified the following passage of the manuscript accordingly:

“As the theoretical lower bound for $\sigma_{\hat{N}}$, we considered it equal to $\sqrt{\hat{N}}$, which arises from the fact that \hat{N} is Poisson distributed and that in SMLM the photon counts of each fluorophore are typically determined in one single measurement. We note, however, that in real-life experiments, other factors may enlarge this value. For example, the variance introduced by EM amplification in EM-CCD cameras used in SMLM can lead to errors in photon counts that are a factor of 2 larger than Poisson statistics⁵⁴. Also, the presence of a faster blinking process occurring with rates comparable to the camera frame rate may increase the variability of \hat{N} .”

Minor:

Comment (I-8): - *“invariability of the shape of the single molecule images”; formulated in more conventional terms, I imagine you are referring to the effective lateral point-spread function being close to constant over the thickness of the usable depth (~250 nm). This seems expected, the additional impact of the illumination function is a scaling that due to the stratified nature of the illumination field is simply an intensity scaling. It might be useful to discuss in these or similar terms.*

Response (1-8): Following Reviewer #1 suggestion, we have modified the manuscript to denominate the shape of single molecule signals also as “effective lateral point-spread function”.

The invariance of the effective lateral PSF of molecules located throughout the TIRF z-range may seem expected to Reviewer #1, but we do not believe it may be obvious to all readers. Especially given the z-dependent variations of the angular emission pattern, that we show and do influence the collected fluorescence signal. For this reason, we decided to show it explicitly.

Comment (1-9): “in good agreement with what is expected for an immunolabeled microtubule (primary antibodies and Fab fragments from secondary antibodies).” Please provide suitable references.

Response (1-9): We thank the reviewer for this comment. We have added the following suitable reference which provides estimates for the diameter of immunolabeled microtubules using different labelling strategies: Nat. Methods, 9, 582-584 (2012).

Comment (1-10): On pg 14: “As an example, Figure 2d shows the normalized profiles of average signals...” this seems to refer to Fig. 2e.

Response (1-10): We appreciate the reviewer for pointing this mistake out. We have corrected the reference to that Figure in the revised version of the manuscript. We note that Figure 2c was moved to Supplementary Figure 1d. So, the original Figure 2e is now Figure 2d.

Reviewer #2 (Remarks to the Author):

In this manuscript Szalai et al report on a methodology based on total internal reflection microscopy (TIRF) to obtain sub-10nm axial nanometric resolution over a depth of around 250nm from the glass interface, in combination with single molecule localization DNA-PAINT or dSTORM. The method is in fact extremely simply and already inherent to current SMLM set-ups, thus straightforward to implement. It is very well known in the community that by proper calibration of the evanescent field in the z-direction one could determine the axial position of an object with higher resolution than that fixed by diffraction. Yet, the authors do a proper job by making a quantitative study on the dependence of the evanescent field in the axial direction, as a function of several parameters and combine it with the concepts of SML to obtain nearly isotropic nanometre resolution in 3D. Of particular relevance, the authors consider in their model the angular emission of individual molecules to account for variations in the detected fluorescence signal. The manuscript is well written, the data are solid and importantly in my opinion, the simplicity of this method makes it useful to the community.

There are pro's and con's of the method: The major advantages rely on the simplicity, since it does not require a different set-up, neither adaptation of a current SMLM, i.e., basically all the SMLM methods work under TIRF illumination. The resolution is superior that most commonly used methods (astigmatism or DONALD) and comparable to that of iPALM or iSTORM, but orders of magnitude simpler to implement. On the other hand, it bears an inherent limitation: the fact that the method is essentially restricted to axial distances below than 250nm (the penetration depth of the evanescent field). Considering recent efforts in the field to achieve comparable axial nm resolution deeper into the sample and/or to extend the axial working range to several microns, the excitement for this work by the community might be modest.

I have a few remarks that would need to be addressed in a revised version:

Comment (2-1): *1) In the first part of the paper, the authors rightly highlight the importance of including the angular emission pattern of individual dipoles in the calculation of the collected fluorescence intensity. Indeed in Figure 1c,d, they estimate the emission pattern of molecules according to their dipole orientation with respect to the glass-water interface. However, the equations that follow later and used in the ms to obtain the parameters needed for determining axial positions, only consider an average dipole emission, assuming that fluorophores will rotate freely during the time scale of the experiments. I am not fully understand why they make this assumption. Most super-resolution experiments are performed on fixed samples, where most probably the fluorophores are not freely rotating and therefore having different fixed dipole emissions. Thus, either the authors experimentally demonstrate free rotation of fluorophores on fixed samples so that their equations are valid, or, would need to adapt their model to include the influence of dipole emission. In the latter case, I guess they would need to have prior knowledge of the dipole emission of each detected molecule, which would add complexity to the set-up.*

Response (2-1): We thank Reviewer #2 for raising this point. The assumption of freely rotating fluorophores is actually very usual in SMLM. The great majority of SMLM methods, for example all 2D SMLM measurements that use Gaussian fits to determine the position of the molecules, work under this assumption.

More importantly, the specific configurations used in our paper (DNA-PAINT with Atto-655 and dSTORM with antibodies conjugated to AlexaFluor647) have been widely used in SMLM measurements and deliver accurate images through localization using Gaussian fits (e.g. using popular analysis software like Thunderstorm or Picasso), which demonstrates the validity of the freely rotating fluorophore approximation for these configurations.

We may add as an explanation that even though the biological samples are fixed, this procedure does not impede the motion and diffusion of small molecules. In fact, the fluorescent immunolabeling takes place after fixation and the fluorophores are linked to the antibodies through flexible bonds, meaning that they can rotate during the measurement. In the case of DNA-PAINT, Atto655 is linked to oligonucleotides which are freely diffusing in the solution, and through the fixed sample, before they bind to the complementary oligonucleotide coupled to the antibody. In the case of dSTORM measurements, Alexa647 is linked to the antibodies through a single chemical bond.

We have clarified this point now in the manuscript with the following passage:

“In addition, the isotropic average (CF_{avg}) is also shown, which corresponds to the usual experimental situation in SMLM methods where fluorophores, bound to antibodies (i.e. dSTORM) or oligonucleotides (i.e. DNA-PAINT) via flexible bonds, can rotate during the measurement time.”

Comment (2-2): 2) *The authors argue that the method is easy to implement and provide a look up table for users. Nevertheless, the accuracy of the method will ultimately depend on the quality of the set-up and calibration might not always be easier & comparable from one set-up to another. I suggest that for completeness the authors add a paragraph on the discussion regarding a word of caution to users and how critical the method is to inaccuracies of the set-up and/or the prior parameters to be determined.*

Response (2-2): We thank Reviewer #2 for this comment, though we must disagree. Instead of adding a word of caution for potential users, we would like to strongly encourage anyone interested to adopt SIMPLER because it is an easily implementable and reliable method. That said, we are of course happy to further explain how critical or not are the experimental parameters on the end performance of SIMPLER.

The main two experimental parameters that affect the overall accuracy of the axial localization are the angle of incidence, θ_i and the number of photons at $z = 0$, N_0 . Both parameters can be determined with sufficient accuracy by performing independent measurements.

Determination of N_0 requires a single SMLM measurement, and therefore should be doable by any user. Determination of θ_i , is the only “new” calibration needed for a SMLM user. We demonstrate the performance of SIMPLER on two different microscopes, where θ_i was determined in two different ways. In our home-built set-up, we used the lateral displacement method (Opt. Express 2018, 26: 20492-20506) as described in Supplementary Method 1. The commercial microscope (Nikon NSTORM 5.0) counts with a calibrated motorized stage to vary the angle of incidence; we applied SIMPLER with the value of θ_i according to the instrument calibration. In both cases, SIMPLER delivered perfect flawless 3D images from the first attempt. We also would like to point out that the two types of measurements were performed in different laboratories by different persons.

Nevertheless, we take this point raised by Referee #2, also in Comment 2-4 in a more quantitative way. In the new version of the manuscript, we provide new Supplementary Figures 1 (Figure R3), 5 (Figure R4), and 6 (Figure R5), that illustrate the robustness of SIMPLER by showing the magnitude of axial distortions generated when using incorrect values of N_0 , θ_i or α . We refer Referee #2 to our Response 2-4 below.

In addition, to further facilitate the implementation and testing of SIMPLER, in the new version of the manuscript we provide a Supplementary Software with an intuitive graphical user interface and example data.

Comment (2-3): 3) Figure 1g: what are the different lines (and colours) associated to SIMPLER? Very difficult to see & distinguish the different colours. The text associated to this figure (page 9) is also very confusing: comparison between d_F & N_0 for different axial lengths... how do these numbers relate to the figure?

Response (2-3): We thank Reviewer #2 for this comment. We acknowledge that Figure 1g was sub-optimal. We have improved Figure 1g (see Figure R6) and clarified its description in the manuscript. We have expanded the description of Figure 1g to read as follows:

Figure R6 (Figure 1g). (g) Theoretical lower bound for the axial localization precision of SIMPLER (equation 3) for different sets of N_0 and σ_{d_F} with $d_F = 87.5$ nm and $\alpha_F = 0.93$. Comparison of the theoretical localization precision of SIMPLER with respect to the reported precision of two other well-established z -localization techniques: single lens astigmatism and DONALD, for $\hat{N}_0 = 7,000$ photons (data taken from¹⁰).

“Figure 1g displays curves of σ_z as a function of the axial position computed with equation 3, taking $\sigma_{\hat{N}} = \sqrt{\hat{N}}$ for different experimentally accessible values of N_0 and σ_{d_F} (d_F and α_F were fixed to 87.5 nm and 0.93, respectively). N_0 was varied between 5,000 and 50,000 photons per frame to account for a wide range of experimental conditions (different laser powers, frame rates, fluorophores, etc). On the other hand, σ_{d_F} was ranged from 1 nm to 5 nm as these values relate to sensible extreme limits for the standard error of θ_i (0.3° to 1.5°). This shows that under usual experimental conditions, SIMPLER is potentially able to deliver axial localization precisions of just a few nanometres.”

Also, we revised all related passages of the manuscript to make clear that the theoretical curves σ_z vs. z were calculated with equation 3.

Comment (2-4): 4) Page 11: Explanation of Figure 2c in the text is also confusing: where does the 10nm average accumulated distortions within 0-250 axial range comes from?

Response (2-4): We acknowledge Reviewer #2 that this passage of the manuscript was not clear enough. Reviewers #1 and #3 have made comments related to this part of the manuscript. In response to these comments, we have re-written this part of the manuscript.

Figure 2c is now incorporated in Supplementary Figure 1, which together with the new Supplementary Figures 5 and 6, describe all possible axial mislocalizations or distortions that can be generated by using incorrect values for the calibration parameters. We refer Reviewer #2 to our Response (1-6).

Comment (2-5): 5) Page 11: last sentence: what do the authors mean by “almost fixed xy position”.????

Response (2-5): This was meant to indicate the value for single-molecules. We believe this now clear in the revised version of the manuscript, after the modifications made in response to the previous Comment (2-4), and related comments by the other Reviewers.

Comment (2-6): 6) Figure 3d: The caption is not clear enough.

Response (2-6): We appreciate Reviewer #2 for pointing this out. We have improved the caption of Figure 3d as follows:

“(d) Histogram of the experimentally determined variance of \hat{N} ($\sigma_{\hat{N}}$), expressed in units of $\sqrt{\hat{N}}$ to remark how much larger is $\sigma_{\hat{N}}$ with respect to the expected theoretical lower bound ($\sigma_{\hat{N}} = \sqrt{\hat{N}}$). The values 2, 5, and 8 are highlighted.”

Comment (2-7): Page 17: What do they mean by the “complete working range”? Are the authors referring to 0-250nm axial range?

Response (2-7): Yes, exactly. We have clarified this explicitly in the revised version of the manuscript.

Reviewer #3 (Remarks to the Author):

The manuscript by Szalai, et al. describe a new approach called SIMPLER used to estimate the axial position of single molecules in dSTORM when performed using TIRF imaging. SIMPLER relies on the intensity of the single molecules showing a dependence on the z-position in the evanescent wave. Using the intensity of molecules at the coverglass as a reference, the signals can be converted into axial positions and plotted in three dimensions. The reported axial resolutions, which can be achieved on relatively simple

or at least commercially available microscopes using existing image analysis software, rival or best some of the most precise single molecule methods based on more advanced instrumentation. The resulting data and final images, such as the cross-sections of microtubules, are compelling and suggest the authors have developed an approach which can be adopted by almost any scientist with access to a TIRF microscope. However, I am unclear about how the uncertainties in the axial positions of the molecules are determined and displayed for several examples. I am also unclear why the uncertainty for some parameters are not factored into the final axial position uncertainty. Given the potential power and widespread use of this approach and that the precision of localization is arguably one of the most critical results provided by SIMPLER, adequate answers or well-defined arguments to the points listed below will hopefully clarify some of the key features of this method.

Comment (3-1): *1- The photons per molecule per frame in figure 2b shows a much larger mean and a much smaller distribution than other investigators have typically measured for single molecules of Alexa647 and ATTO655. Is this simply due to the selection of molecules which last 3 or more frames? It would be helpful if the authors showed examples of unfiltered histograms. This might also help readers appreciate how many molecules are excluded from the final images and the authors should provide this information for each figure. Moreover, plotting a few of the images with all of the molecules would provide a helpful comparison.*

Response (3-1): We have expressed single-molecule emission intensity in photons/frame. Taking this into account, the distribution of photon counts shown in Figure 2b is in line with what other researchers have observed. Take for example the distribution of photons/second for Alexa 647 reported by Nieuwenhuizen, M. Bates et al. (Plos One, 2015, [10.1371/journal.pone.0127989](https://doi.org/10.1371/journal.pone.0127989) - Fig. S6). They show a distribution centered at around $(1.75 \pm 0.2) \times 10^5$ photon/s, while our mean photon count rate, expressed in photons/s is $\sim (2.04 \pm 0.2) \times 10^5$ (considering our camera frame time of 250 ms). Both values are really similar, especially taking into account that the exact value of the detected photon count rate depends on laser irradiance, detection filter set, and quantum efficiency of the camera.

Of course, we acknowledge Reviewer #3 that comparison to previous works should be facilitated. Therefore, in the revised version of the manuscript, we have included the frame duration in the legend of the revised Figure 2b. Also, as suggested by Referee #3, we are including the number of molecules used in each figure, together with the total number of molecules accounted before the filtering step.

With respect to how many molecules (or better said, localizations) are discarded through the frame-filtering step, this depends on the acquisition conditions and SMLM method (e.g. dSTORM or DNA-PAINT). For example, if the acquisition is set to detect each single molecule, in average, during 5 frames. Then, discarding 2 frames (first and last) corresponds to discarding 40% of the localizations. Naturally, this fraction is reduced proportionally as the single-molecule emission events are detected over more frames.

In the case of DNA-PAINT measurements, the frame filtering has the additional advantage that also serves to exclude short emission events due to nonspecific binding of the labelled oligonucleotides. This is a well-known fact, reported for example in Stein et al. *Nano Lett.* 2019, 19, 11, 8182–8190. We reproduce the distribution of photon counts from the supporting information of that paper in Figure R7.

Figure R7. DNA-PAINT image (a, c) and photon counts histograms (b, d) of localisation events observed in a surface-passivated sample containing (a) no DNA-origami or (b) DNA-origami structures both imaged with 10 nM Cy3b-labeled DNA imager strands. Figure adapted from *Nano Lett.* 2019, 19, 11, 8182–8190.

We note that the non-specific binding of the labelled oligonucleotides depends on the particular DNA sequence, fluorophore and sample. We have now included a Supplementary Figure 3 (Figure R8) where we show the histogram from Fig. 2b together with the raw distribution of N_0 . From the comparison of the two histograms, the effectiveness of the frame filtering to remove the non-specific binding population is evident.

Figure R8 (Supplementary Figure 3). Example photon count histograms of a DNA-PAINT experiment, before and after frame filtering. (a) Histogram of photon counts per frame (frame time 250 ms) for all localizations detected in the sample for the determination of \bar{N}_0 (total localizations: 21,639). (b) Histogram of photon counts per frame for the valid localizations after frame filtering (total valid localizations: 5,030). The frame filtering procedure eliminates low count frames originated from i) specific DNA binding events that lasted less than three frames, and ii) shorter events due to non-specific binding.

We have modified the manuscript text to make this point clearer and to make a reference to Supplementary Figure 3 (Figure R8) for comparison:

“This frame-filtering rules out low-intensity events that would bias axial localizations to artificially higher z-values. In the case of DNA-PAINT measurements, the frame-filtering has the additional advantage of excluding short emission events due to nonspecific binding of the labelled oligonucleotides.”

“The distribution of \widehat{N}_0 (after the correction for local intensity and frame filtering) is well described by a normal distribution with an average value of 51,000 photons and a standard deviation of 10% (For comparison, Supplementary Figure 3 shows the non-frame filtered photon count histogram).”

Comment (3-2): *2- How are the molecules plotted? From the methods section, I understand that a 2nm Gaussian blur was applied to all directions. This implies a precision of 2nm or less, but this is not the axial resolution described throughout the full TIRF range in the text. Moreover, results presented in figure 3e indicate that plotting with a Gaussian blur should be performed with a much larger sigma.*

Response (3-2): We thank Reviewer #3 for raising this point, and the related Comments 3-3, 3-4 and 3-5, as it allows us to further clarify how we analysed the single-molecule localization data.

There are basically two ways to construct a super-resolved image from SMLM data, each one with pros and cons. One way, the one mentioned by Reviewer #3, consists of plotting each molecule as a Gaussian with standard deviation (SD) equal to the experimentally determined localization error. This method has the advantage that images are rendered with a clear statistical criterium. The disadvantage is that, in order to obtain an experimental measure for the localization error, it is necessary to merge consecutive localizations using some prior assumption about distance between localizations. Alternatively, one can plot all valid localizations. In this case, multiple localizations are plotted that may correspond to the same fluorophore (dSTORM) or the same binding site (DNA-PAINT). This approach has the advantage that it does not require any prior assumption and all the data is plotted. The disadvantage is that each labelled spot appears blurred due to the multiple localizations with slightly different positions. For this reason, when using the latter approach, it has become usual to take a smaller SD to render images; see for example Nature Methods, 16, 387-395, 2019; Nature Methods, 16, 1045-1053, 2019, where a standard deviation of 0.4 of the average standard error of localization is used.

We have chosen to plot all valid localizations. Super-resolution images were then constructed by rendering each localization as a 2D Gaussian peak with a predefined standard deviation (SD) proportional (in our case we took a factor of 0.5) to the average localization precision. Regrettably, the information about the SD used for the Gaussian rendering was incomplete in the Methods section. We apologize for that and thank Reviewer #3 for asking about this point. The value of SD = 2 nm was used to render DNA-PAINT data (Fig. 2d, Fig. 3a-b, Fig S6, Fig. S8-S10). dSTORM data was rendered with SD = 6 nm (Figure 4a-b, Fig S12). We have improved the Methods section to explain that images were rendered using all valid localizations and rectified the information about the SD values used for dSTORM and DNA-PAINT:

“Finally, z-color-coded image rendering was done using the ImageJ plug-in ThunderStorm⁶³, importing the list of all (x, y, z) coordinates without merging localizations re-appearing in subsequent frames. A Gaussian filter with a size proportional (factor 0.5) to the median localization precision in the 0-250 nm axial range (sigma = 2 nm, for DNA-PAINT data and sigma = 6 nm for dSTORM data) was used for all three dimensions.”

Comment (3-3):3- Can the authors also clarify how the lateral and axial positions for molecules lasting more than 3 frames are determined. Are the positions determined from the average of the positions in each frame? From the average of the photons per frame for axial? Summing the photons across frames for lateral?

Response (3-3): Following on Response 3-2, it is clear now that we have chosen to render images using all valid localizations, not molecules. That means, we determined an (x, y, z) position for all single-molecule emission events detected during a complete camera frame. Lateral (x, y) positions were obtained through a Gaussian fit using any 2D SMLM software (Picasso in our case). Axial (z) positions were obtained through SIMPLER. This part is, we believe, is clear in the manuscript.

In the revised version of the manuscript, we state this explicitly and refer to the Methods section for further details:

“We reconstructed super-resolved images by plotting all valid localizations after frame-filtering (further details in the Methods section).”

We have also improved the explanation about the frame filtering step in the Methods section:

“To ensure the molecule emitted during the whole exposure time, localizations were kept as valid only in the case that other localizations, reasonably attributed to the same fluorophore (within a 20 nm), were detected in the previous and subsequent frames (Figure 2a). Therefore, a localization (x_n, y_n) found in the frame n was kept as valid only if there was another localization in frame $n - 1$ located at $[(x_n - x_{n-1})^2 + (y_n - y_{n-1})^2] < (20 \text{ nm})^2$ and another localization in frame $n + 1$ located at $[(x_n - x_{n+1})^2 + (y_n - y_{n+1})^2] < (20 \text{ nm})^2$. Molecules detected for less than three frames were thus ignored.”

Additionally, in the revised version of our manuscript, we are providing a Supplementary Software to perform SIMPLER from a list of single-molecule localizations obtained from most popular SMLM analysis software. The software, which includes an intuitive graphical user interface and example data, can perform the filtering step described above.

Comment (3-4): 4- In the comparison of filtered and unfiltered data in supplementary figure 5, the number of molecules should remain constant while the molecule positions in the unfiltered should simply be shifted to a higher axial position due to a decrease in their average number of photons per frame. However, the unfiltered data images seem to show a lot more molecules than the filtered data images. In both examples, it is unclear from where in the filtered image the molecules are being axially translated when plotted in the unfiltered images.

Response (3-4): Now that we have clarified that we construct the super-resolved images using all localizations (not molecules), we believe it is clear why the unfiltered images show more localizations.

Comment (3-5): 5- Clarification on the estimation of the variances used will also be helpful. Most of the points below concern this critical part of the method. Can the authors please define the variances estimated and used in the error determination for each of the figures in which example images are plotted?

Response (3-5): We thank Reviewer #3 for this and the following comments, as it allows us to further clarify on the axial localization precision. We infer from this and subsequent Comments (especially Comment 3-13) that Reviewer # 3 has interpreted that SIMPLER requires an estimation of the axial localization precision based on prior knowledge of the variances of the parameters involved. However, this is not the case. The axial localization precision (σ_z) shown in Figures 3e and 4d are determined experimentally as the standard deviation of the z coordinate determined for the same single molecule emitting in subsequent frames.

The calculations of σ_z through equation 3 are used only to provide a theoretical framework, a lower bound for the axial localization error, with the aim of gaining insight into the contributions of the different parameters to the experimental precision.

We thank the reviewer for this question as it highlights that we need to provide a clearer explanation in the manuscript about the experimental determination of the axial localisation error. We have modified the corresponding passage of the paper, where we first discuss experimental determinations of σ_z , as follows:

“To obtain insight into the critical parameters for SIMPLER, we determined experimentally the axial localization precision (σ_z) and compared it to the theoretical predictions (equation 3). From the same SIMPLER – DNA-PAINT experiment ($\hat{N}_0 = 51,000$), single-molecule emission events longer than 5 camera frames were selected to compute an experimental measure of σ_z . In this way, after filtering out the first and last frames (Figure 2a), at least three independent measurements of \hat{N} , and their corresponding estimations of \hat{z} , were available for every single molecule. Figure 3c shows the obtained distributions of experimental σ_z (extracted from 366 single-molecule traces) grouped for different ranges of z .”

In addition, at the end of that discussion, we have added the following sentence to make it absolutely clear that SIMPLER does not require any priors about the variances of any parameter:

“We remark though that this analysis is only performed to validate the applicability of equation 3. It is not necessary for the application of SIMPLER, as its implementation does not require any prior knowledge about the variances of the parameters involved.”

And in the Method section:

“Experimental measures of the z -localization precision (σ_z) were determined from the variance of z -localizations of the same molecule. Values of σ_z were registered for multiple molecules located at different z -positions as described in the main text (Figures 3e and 4d).”

Comment (3-6): 6- I do not understand the results in Figure 1e which is supported by Supplementary figure 1a. From my reading of the text, the exact solution represents the product of the red line CF in figure 1d and the evanescent wave approximation (the blue line in figure 1e). When normalized at $z=0$, shouldn't the

exact solution always display a steeper decline and a lower amplitude than the evanescent wave approximation at all values $z > 0$?

Response (3-6): We thank Reviewer #3 for this comment. It is just a confusion. The exponential curve shown in Figure 1e is not the evanescent excitation field, but a fit to the exact solution.

$I(z)$ is the evanescent excitation field (Figure 1b). The detected fluorescence signal $F(z)$ will be proportional to the product of the excitation field and the collected fluorescence: $F(z) = I(z) \times CF_{avg}(z)$. This is the exact solution, as it contains the numerically calculated collected fluorescence. It turns out that the exact solution (red curve in Figure 1e) is very well represented by an exponential function (blue curve in figure 1e). This comment made us note that the blue curve displayed in Figure 1e was mistakenly labeled as $I(z)$. We have now corrected it.

Comment (3-7): *7- I am unconvinced that the uncertainty for the $N0$ value is of minor importance as indicated in the text. Could the authors elaborate and help clarify their argument for this conclusion? I understand from the Figure 2c and supplementary figure 2 that choosing a value of $N0$ that does not correspond to $z=0$ will lead to an offset in the position of the other molecules. But shouldn't this be plotted as an uncertainty in the position of the molecules rather than an offset? The histogram in figure 2b should provide σ_{N0} , correct?*

Response (3-7): Reviewer #2 also raised this point in Comment 2-4. Evidently, we were not clear on this passage of our manuscript. We refer Reviewer #3 to our Response 2-4. We believe that, with the improved explanation and new Supplementary Figures 1, 5, and 6, we have cleared all doubts about the robustness of SIMPLER and the influence of each parameter in its end performance.

Comment (3-8): *8- The authors indicate that the lower bound for the uncertainty might be estimated from the square root of the number photons. Figure 2b should offer an adequate test for this approximation. How well does the data in figure 2b approximate this theoretical lower bound? If the approximation is insufficient, the authors should use the uncertainty of the reference molecules located at $z=0$ as the practical lower bound.*

Response (3-8): We do indicate $\sigma_N = \sqrt{N}$ is the theoretical lower bound. But we also say that “in real life experiments, other factors may enlarge this value. For example, the variance introduced by EM amplification in EM-CCD cameras used in SMLM can lead to errors in photon counts that are a factor of 2 larger than Poisson statistics⁵⁴.”

In the revised version of the manuscript, we have improved this paragraph following a suggestion by Reviewer #1 (Comment 1-7), that micro-blinking could also contribute to enlarge σ_N . The new paragraph reads as follows:

“We note, however, that in real-life experiments, other factors may enlarge this value. For example, the variance introduced by EM amplification in EM-CCD cameras used in SMLM can lead to errors in photon

counts that are a factor of 2 larger than Poisson statistics⁵⁴. Also, the presence of a faster blinking process occurring with rates comparable to the camera frame rate may increase the variability of N.”

The best experimental test for σ_N is to analyze it for individual molecules, as we do for example in Figures 3d and 3e, and 4d. The histogram of photon counts at $z=0$ shown in Figure 2b is not a good test for σ_N because it corresponds to molecules distributed over a very large area of $16 \times 16 \mu\text{m}^2$. The width of this distribution is enlarged by other factors, such as imperfect corrections of the excitation intensity and even the size of the antibodies.

Comment (3-9): 9- From analyzing single molecule traces, the authors conclude that the photon uncertainty for molecules $z>0$ can be estimated from $5\sqrt{N}$. Does this hold for N_0 as well?

Response (3-9): We find that in our experiments, the average value of $\sigma_{\hat{N}}$ for single molecules, independently of their z coordinate, is of about $5\sqrt{N}$. This would hold for single molecules located at $z = 0$. However, the distribution of \hat{N}_0 from an ensemble of single-molecule distributed over a large field of view presents a larger variance, for the reasons explained in our previous answer (Response 3-8). In fact, the histogram of \hat{N}_0 that we show in Figure 2b is an example of this. It presents a mean value of 51,000 counts and a standard deviation of 10%. That is a standard deviation of 5,100 counts, $\sim 22 \times \sqrt{N}$.

Comment (3-10): 10- Some of the simulations (such as Figure 1g) indicate the axial uncertainty approaches zero as the axial position approaches $z=0$. Given that the authors find in figure 3 that $5\sqrt{N}$ seems to offer a good approximation for axial uncertainty, it is unclear how it can approach zero even at $z=0$. Can the authors clarify or explain this?

Response (3-10): There seems to be a confusion. The value of $5\sqrt{\hat{N}}$ is a good approximation for $\sigma_{\hat{N}}$, not for σ_z .

The curves of σ_z vs. z shown in Figure 1g are representations of equation (3) for different values of the N_0 and σ_{d_F} . We have modified the caption of Figure 1g to make this clearer:

“(g) Theoretical lower bound for the axial localization precision of SIMPLER (equation 3) for different sets of N_0 and σ_{d_F} with $d_F = 87.5 \text{ nm}$ and $\alpha_F = 0.93$.”

Therefore, although very small at $z = 0$, σ_z can never take the value of zero (equation 3).

Nonetheless, in response to Comment (2-3) by Reviewer #2, we have improved Figure 1g for clarity in the new version of the manuscript (please see Response 2-3).

Comment (3-11): 11- The authors simulate the expected uncertainty for many of the components in equation 3 used to determine the axial positions (Fig. 1g, Fig. 2c, Supp. Fig 1, Supp. Fig. 2). Generally,

the components of interest in each figure introduce <10nm uncertainty over the axial range, but it is unclear what uncertainties are assumed for the other components when making these simulations. It will be helpful if the authors include this information in the legend for each figure?

Response (3-11): In all cases where we show calculated σ_z curves (Fig. 1g, Fig. 3e, and Fig. 4d) we did it using equation (3), which expresses σ_z as a function of $\sigma_{\hat{N}}$ and σ_{d_F} , and neglects $\sigma_{\hat{N}_0}$ and σ_{α_F} . In response to Comments (1-6) and (2-4) of the other Reviewers, we have included new Supplementary Figures 1, 5, and 6 to illustrate the influence of the involved parameters in the determination of axial positions.

We remark again that the performance of SIMPLER does not depend on any theoretical estimation of the variances of the involved parameters. We compute the theoretical lower bound through equation 3 and compare it to the experimentally determined σ_z (Figs. 3e and 4d) just to gain and provide insight into the critical parameters and sources of error. We have added the following sentences to make this clearer in the new version of the manuscript:

“We remark though that this analysis is only performed to validate the applicability of equation 3. It is not necessary for the application of SIMPLER, as its implementation does not require any prior knowledge about the variances of the parameters involved.”

Comment (3-12): *12- The parameters d_f and α_f are derived from a fit of $I(z) \times CF(z)$, both of which are calculated based on instrument and sample parameters. The use of just these calculated values is worrisome. It would be more compelling if the authors offered experimental determinations for the values using their imaging systems to compare with the simulated values.*

Response (3-12): We thank Reviewer #3 for this Comment. While it would be nice to measure $I(z)$ and $CF(z)$ independently, such measurements are not trivial and would imply experimental conditions different from the real biological samples, which in turn would call for approximations and calculations to adjust the measurement results to the new conditions. In summary, that is unfortunately not a viable approach. That said, we would like to remark that the end performance of SIMPLER is also a way to validate the values of d_F and α_F . Using structures of known and fixed geometry as standards, such as the microtubule cross-sections or the nuclear pore complex, it is straightforward to validate the values of d_F and α_F (and of N_0). Any mismatch in those values leads to distinctive distortions in the images, as summarized in Supplementary Table 2 and explained in more detail in the new Supplementary Figures 1, 5, and 6; please see Response 2-4.

In addition, in the new version of the manuscript, we provide a Supplementary Software to apply SIMPLER from a list of localizations. The software includes an intuitive graphical user interface and a specific module to test and adjust the values of θ_i , α and N_0 , using the retrieved SIMPLER image as feedback (an example of the output of this module is illustrated in the new Supplementary Figure 7).

We also mention this in the revised version of the main manuscript text:

“To ease the implementation of SIMPLER to the wide-imaging community, we also make available a Supplementary Software with an intuitive graphical user interface (GUI) that directly outputs z-position from 2D SMLM analysis lists (Supplementary Software 1). The software computes $F(z) = I(z) \times CF_{avg}(z)$ and provides d_F and α_F for each user input experimental conditions: NA ; λ_0 ; λ ; θ_i ; n_i ; n_s and α . Among other features, the software can correct photon counts for uneven illumination and automatically perform the frame filtering step. In addition, taking advantage of the distinctive effects of each of the calibration parameters θ_i , α and N_0 , (Supplementary Table 2, Supplementary Figures 1, 5 and 6) their values can be easily adjusted using 3D images of reference structures as feedback. The Supplementary Software includes a specific module for this purpose and, as illustrated in Supplementary Fig. 7, it is extremely useful to find the best estimate for a parameter that has been determined or estimated with low accuracy.”

Comment (3-13): *13- It is unclear how the uncertainty for the TIRF depth, σ_{d_F} , and the uncertainty for the non-evanescent component, σ_{α_F} , are estimated in each experiment. Given they are required to calculate the axial position uncertainty, good estimates for these values are likely critical. The authors have included simulations for these which indicate they have little bearing on the axial position uncertainty. However, it is unclear if those simulated uncertainties reflect uncertainties in an experimental setting. Could the authors please elaborate on the estimation of these uncertainties?*

Response (3-13): The uncertainties of σ_{d_F} and σ_{α_F} are not estimated for each experiment. That is not necessary to implement SIMPLER, nor is the a priori estimation of any variance of the other involved parameters.

What we do is to determine experimentally the value of σ_z from direct measurements, without any prior calculation, from the variance in multiple z-localizations of single-molecules located at different z-positions (Figures 3e and 4d). Then, we compare the experimentally determined values of σ_z vs. z to the calculated uncertainties through equation 3, using sensible ranges of σ_{d_F} and σ_{α_F} with the only purpose of gaining insight into the individual contributions to the final localization error. We remark this is not necessary for the application of SIMPLER. It is just a way to gain an understanding on the influence of the different parameters. In the revised version of the manuscript, this is done in much more detail. This issue was addressed also in our Response (1-6) to Reviewer #1 and Response (2-4) to Reviewer #2.

Minor points

Comment (3-14):• *Page 22; sentence 2. Typo.*

Response (3-14): We thank the reviewer for pointing out this typo. We have corrected it in the current version of the manuscript.

Comment (3-15):• *For one of the super-resolution microscopes, a sCMOS camera was used in the single molecule localization experiments, but it was unclear from the methods if the corrections for the pixel dependent noise common in these cameras was applied. Given the noise characteristics for these cameras, this correction is usually required for single molecule imaging. If they authors performed these corrections, please indicate this in the methods. If not, the authors may wish to revisit data from images produced by this instrument.*

Response (3-15): We thank Reviewer #3 for noticing this missing information. We used the intrinsic hot pixel correction of the sCMOS ORCA Flash 4.0 V3, Hamamatsu camera. This feature identifies erroneous hot pixels by comparing a pixel to its nearest neighbours. We have now clarified this in the methods section of the manuscript.

REVIEWER COMMENTS

Reviewer #1 (Remarks to the Author):

I am satisfied with the response and believe the revised MS is greatly improved.

I especially liked the software provision and came across 2 small bugs for which I include diffs below.

While I like the software provision and example data a lot I would request that example localisation data covering slightly larger regions of interest are provided, too. A data volume 10x what is provided right now would be entirely acceptable for readers and provide a better overview of the data that can be achieved.

```
--- export_function.m~ 2020-07-11 22:52:00.000000000 +0100
+++ export_function.m 2020-08-16 22:50:13.000000000 +0100
@@ -51,13 +51,13 @@
simpler_output(:,3)>y1(1) & simpler_output(:,3)<y1(2));
end
x1 = simpler_output(c1,1);
- y1 = simpler_output(c1,1);
+ y1 = simpler_output(c1,2);
z = simpler_output(c1,3);
exportation_file_xyz(x1,y1,z,filename_wformat,handles);
elseif rz_xyz == 4
c2 = 1:size(simpler_output,1);
x1 = simpler_output(c2,1);
- y1 = simpler_output(c2,1);
+ y1 = simpler_output(c2,2);
z = simpler_output(c2,3);
exportation_file_xyz(x1,y1,z,filename_wformat,handles);
end
```

Reviewer #2 (Remarks to the Author):

The authors have candidly addressed all the concerns of the reviewers, in particular mine. They have included new supplementary data to support some of their initial unclear statements, improved the explanation regarding some figures and included some new sentences that address my initial concerns. I have no hesitation to recommend this paper for Nature Comm.

Reviewer #3 (Remarks to the Author):

The manuscript by Szalai, et al. describe a new approach called SIMPLER used to estimate the axial position of single molecules in dSTORM when performed using TIRF imaging. SIMPLER relies on the intensity of the single molecules showing a dependence on the z-position in the evanescent wave. Using the intensity of molecules at the coverglass as a reference, the signals can be converted into axial positions and plotted in three dimensions. The reported axial resolutions, which can be achieved on relatively simple or at least commercially available microscopes using existing image analysis software, rival or best some of the most precise single molecule methods based on more advanced instrumentation. The resulting data and final images, such as the cross-sections of microtubules, are compelling and suggest the authors have developed an approach which can be adopted by almost any scientist with access to a TIRF microscope. However, I am unclear about how the uncertainties in the axial positions of the molecules are determined and displayed for several examples. I am also unclear why the uncertainty for some parameters are not factored into the final axial position uncertainty. Given the potential power and widespread use of this approach and that the precision of localization is arguably one of the most critical results provided by SIMPLER, adequate answers or well-defined arguments to the points listed below will hopefully clarify some of the key features of this method.

>>>Szalai et al. have addressed most of comments satisfactorily. My additional comments to the rebuttal are placed in bold type in line below. I am still positive about the paper since it represents a great addition to the super-resolution imaging toolbox. I have just a points on which I am confident the authors can offer clarification or convince me of my misunderstanding. All of points concern the axial localization uncertainty and since it is arguably the most important factor for a localization method, it may benefit the authors to address the simplest and perhaps even naïve questions at this stage rather than after publication.

***Comment (3-1):**1- The photons per molecule per frame in figure 2b shows a much larger mean and a much smaller distribution than other investigators have typically measured for single molecules of Alexa647 and ATTO655. Is this simply due to the selection of molecules which last 3 or more frames? It would be helpful if the authors showed examples of unfiltered histograms. This might also help readers appreciate how many molecules are excluded from the final images and the authors should provide this information for each figure. Moreover, plotting a few of the images with all of the molecules would provide a helpful comparison.*

Response (3-1): We have expressed single-molecule emission intensity in photons/frame. Taking this into account, the distribution of photon counts shown in Figure 2b is in line with what other researchers have observed. Take for example the distribution of photons/second for Alexa 647 reported by Nieuwenhuizen, M. Bates et al. (Plos One, 2015, [10.1371/journal.pone.0127989](https://doi.org/10.1371/journal.pone.0127989) - Fig. S6). They show a distribution centered at around $(1.75 \pm 0.2) \times 10^5$ photon/s, while our mean photon count rate, expressed in photons/s is $\sim (2.04 \pm 0.2) \times 10^5$ (considering our camera frame time of 250 ms). Both values are really similar,

especially taking into account that the exact value of the detected photon count rate depends on laser irradiance, detection filter set, and quantum efficiency of the camera.

>>> I was actually considering the studies in these papers (*Biomedical Optics Express* (2013) 4:885-889; *Nature Methods* (2011) 8:1027-1036), but I actually agree with authors concerning different instruments will likely produce different mean values, etc. The authors could easily argue they are simply better at building machines, sample prep, etc. I'm considering such a photon/frame distribution to be key to the success of SIMPLER. Since the Alexa647 distribution does appear qualitatively and quantitatively different from other literature descriptions, other readers will likely also raise these questions. Anyway, I think response figure R8 (new Fig. S3) addresses the homogeneous distributions.

Of course, we acknowledge Reviewer #3 that comparison to previous works should be facilitated. Therefore, in the revised version of the manuscript, we have included the frame duration in the legend of the revised Figure 2b. Also, as suggested by Referee #3, we are including the number of molecules used in each figure, together with the total number of molecules accounted before the filtering step.

With respect to how many molecules (or better said, localizations) are discarded through the frame-filtering step, this depends on the acquisition conditions and SMLM method (e.g. dSTORM or DNA-PAINT). For example, if the acquisition is set to detect each single molecule, in average, during 5 frames. Then, discarding 2 frames (first and last) corresponds to discarding 40% of the localizations. Naturally, this fraction is reduced proportionally as the single-molecule emission events are detected over more frames.

In the case of DNA-PAINT measurements, the frame filtering has the additional advantage that also serves to exclude short emission events due to nonspecific binding of the labelled oligonucleotides. This is a well-known fact, reported for example in Stein et al. *Nano Lett.* 2019, 19, 11, 8182–8190. We reproduce the distribution of photon counts from the supporting information of that paper in Figure R7.

Figure R7. DNA-PAINT image (a, c) and photon counts histograms (b, d) of localisation events observed in a surface-passivated sample containing (a) no DNA-origami or (b) DNA-origami structures both imaged with 10 nM Cy3b-labeled DNA imager strands. Figure adapted from *Nano Lett.* 2019, 19, 11, 8182–8190.

We note that the non-specific binding of the labelled oligonucleotides depends on the particular DNA sequence, fluorophore and sample. We have now included a Supplementary Figure 3 (Figure R8) where we show the histogram from Fig. 2b together with the raw distribution of N_0 . From the comparison of the two histograms, the effectiveness of the frame filtering to remove the non-specific binding population is evident.

Figure R8 (Supplementary Figure 3). Example photon count histograms of a DNA-PAINT experiment, before and after frame filtering. (a) Histogram of photon counts per frame (frame time 250 ms) for all localizations detected in the sample for the determination of \widehat{N}_0 (total localizations: 21,639). **(b)** Histogram of photon counts per frame for the valid localizations after frame filtering (total valid localizations: 5,030). The frame filtering procedure eliminates low count frames originated from i) specific DNA binding events that lasted less than three frames, and ii) shorter events due to non-specific binding.

We have modified the manuscript text to make this point clearer and to make a reference to Supplementary Figure 3 (Figure R8) for comparison:

“This frame-filtering rules out low-intensity events that would bias axial localizations to artificially higher z-values. In the case of DNA-PAINT measurements, the frame-filtering has the additional advantage of excluding short emission events due to nonspecific binding of the labelled oligonucleotides.”

“The distribution of \widehat{N}_0 (after the correction for local intensity and frame filtering) is well described by a normal distribution with an average value of 51,000 photons and a standard deviation of 10% (For comparison, Supplementary Figure 3 shows the non-frame filtered photon count histogram).”

>>> I fear I was unclear with my question. My concern was only in how many molecules were excluded compared to how many were kept for plotting the image. Inclusion of the values as discussed above will suffice.

Comment (3-2): 2- How are the molecules plotted? From the methods section, I understand that a 2nm Gaussian blur was applied to all directions. This implies a precision of 2nm or less, but this is not the axial resolution described throughout the full TIRF range in the text. Moreover, results presented in figure 3e indicate that plotting with a Gaussian blur should be performed with a much larger sigma.

Response (3-2): We thank Reviewer #3 for raising this point, and the related Comments 3-3, 3-4 and 3-5, as it allows us to further clarify how we analysed the single-molecule localization data.

There are basically two ways to construct a super-resolved image from SMLM data, each one with pros and cons. One way, the one mentioned by Reviewer #3, consists of plotting each molecule as a Gaussian with standard deviation (SD) equal to the experimentally determined localization error. This method has the advantage that images are rendered with a clear statistical criterium. The disadvantage is that, in order to obtain an experimental measure for the localization error, it is necessary to merge consecutive localizations using some prior assumption about distance between localizations. Alternatively, one can plot all valid localizations. In this case, multiple localizations are plotted that may correspond to the same fluorophore (dSTORM) or the same binding site (DNA-PAINT). This approach has the advantage that it does not require any prior assumption and all the data is plotted. The disadvantage is that each labelled spot appears blurred due to the multiple localizations with slightly different positions. For this reason, when using the latter approach, it has become usual to take a smaller SD to render images; see for example Nature Methods, 16, 387-395, 2019; Nature Methods, 16, 1045-1053, 2019, where a standard deviation of 0.4 of the average standard error of localization is used.

We have chosen to plot all valid localizations. Super-resolution images were then constructed by rendering each localization as a 2D Gaussian peak with a predefined standard deviation (SD) proportional (in our case we took a factor of 0.5) to the average localization precision. Regrettably, the information about the SD used for the Gaussian rendering was incomplete in the Methods section. We apologize for that and thank Reviewer #3 for asking about this point. The value of SD = 2 nm was used to render DNA-PAINT data (Fig. 2d, Fig. 3a-b, Fig S6, Fig. S8-S10). dSTORM data was rendered with SD = 6 nm (Figure 4a-b, Fig S12). We have improved the Methods section to explain that images were rendered using all valid localizations and rectified the information about the SD values used for dSTORM and DNA-PAINT:

“Finally, z-color-coded image rendering was done using the ImageJ plug-in ThunderStorm⁶³, importing the list of all (x, y, z) coordinates without merging localizations re-appearing in subsequent frames. A Gaussian filter with a size proportional (factor 0.5) to the median localization precision in the 0-250 nm axial range (sigma = 2 nm, for DNA-PAINT data and sigma = 6 nm for dSTORM data) was used for all three dimensions.”

>>> Thanks for this clarification to address my misunderstanding concerning the plots. It raises the issue of plotting localizations instead of molecules. Although it is more or less standard practice in the field, I do not consider this to be a good approach. Plotting at the lower sigma suggests the molecules are better localized than they actually are localized. In addition, under-sampling is often a problem with localization data and can lead to limitations on reaching Nyquist criteria. Thus, plotting the same molecule at slightly different positions has the effect of “connecting the dots” which are often the same dot. I recognize that localization data can be plotted in numerous ways and I’ll respect the authors’ decision to plot their data in this manner and simply inform the reader how it was performed. To that end, inclusion of the statements above will suffice for that purpose.

I urge the authors to reconsider this approach for plotting data in future SIMPLER publications.

Moreover, this particular study has an additional consideration regarding the plotting of molecules. For the lateral localizations, the uncertainty is largely determined for each one based on the number of photons, background, etc. For the axial localizations in the work, the uncertainty is determined by calculating the axial position of the same molecule in multiple frames. Thus, the

determined uncertainty is for the mean position of the molecule rather than the individual localizations. It is not clear that the uncertainty in the mean axial position is applicable to the uncertainty in the individual axial localization positions. If the authors could provide some clarification on this point, it would be helpful for me and would likely be helpful for future users of SIMPLER.

Comment (3-3):3- Can the authors also clarify how the lateral and axial positions for molecules lasting more than 3 frames are determined. Are the positions determined from the average of the positions in each frame? From the average of the photons per frame for axial? Summing the photons across frames for lateral?

Response (3-3): Following on Response 3-2, it is clear now that we have chosen to render images using all valid localizations, not molecules. That means, we determined an (x, y, z) position for all single-molecule emission events detected during a complete camera frame. Lateral (x, y) positions were obtained through a Gaussian fit using any 2D SMLM software (Picasso in our case). Axial (z) positions were obtained through SIMPLER. This part is, we believe, is clear in the manuscript.

In the revised version of the manuscript, we state this explicitly and refer to the Methods section for further details:

“We reconstructed super-resolved images by plotting all valid localizations after frame-filtering (further details in the Methods section).”

We have also improved the explanation about the frame filtering step in the Methods section:

“To ensure the molecule emitted during the whole exposure time, localizations were kept as valid only in the case that other localizations, reasonably attributed to the same fluorophore (within a 20 nm), were detected in the previous and subsequent frames (Figure 2a). Therefore, a localization (x_n, y_n) found in the frame n was kept as valid only if there was another localization in frame $n - 1$ located at $[(x_n - x_{n-1})^2 + (y_n - y_{n-1})^2] < (20 \text{ nm})^2$ and another localization in frame $n + 1$ located at $[(x_n - x_{n+1})^2 + (y_n - y_{n+1})^2] < (20 \text{ nm})^2$. Molecules detected for less than three frames were thus ignored.”

>>> Including rationale for the 20nm cutoff for future users of SIMPLER to set this parameter will be helpful. Otherwise, these clarifications are satisfactory.

Additionally, in the revised version of our manuscript, we are providing a Supplementary Software to perform SIMPLER from a list of single-molecule localizations obtained from most popular SMLM analysis software. The software, which includes an intuitive graphical user interface and example data, can perform the filtering step described above.

Comment (3-4): 4- In the comparison of filtered and unfiltered data in supplementary figure 5, the number of molecules should remain constant while the molecule positions in the unfiltered should simply be shifted to a higher axial position due to a decrease in their average number of photons per frame. However, the unfiltered data images seem to show a lot more molecules than the filtered data images. In both examples, it is unclear from where in the filtered image the molecules are being axially translated when plotted in the unfiltered images.

Response (3-4): Now that we have clarified that we construct the super-resolved images using all localizations (not molecules), we believe it is clear why the unfiltered images show more localizations.

>>> **Responses above clarify the data show localization positions instead of molecule positions.**

Comment (3-5): 5- Clarification on the estimation of the variances used will also be helpful. Most of the points below concern this critical part of the method. Can the authors please define the variances estimated and used in the error determination for each of the figures in which example images are plotted?

Response (3-5): We thank Reviewer #3 for this and the following comments, as it allows us to further clarify on the axial localization precision. We infer from this and subsequent Comments (especially Comment 3-13) that Reviewer # 3 has interpreted that SIMPLER requires an estimation of the axial localization precision based on prior knowledge of the variances of the parameters involved. However, this is not the case. The axial localization precision (σ_z) shown in Figures 3e and 4d are determined experimentally as the standard deviation of the z coordinate determined for the same single molecule emitting in subsequent frames.

The calculations of σ_z through equation 3 are used only to provide a theoretical framework, a lower bound for the axial localization error, with the aim of gaining insight into the contributions of the different parameters to the experimental precision.

We thank the reviewer for this question as it highlights that we need to provide a clearer explanation in the manuscript about the experimental determination of the axial localisation error. We have modified the corresponding passage of the paper, where we first discuss experimental determinations of σ_z , as follows:

“To obtain insight into the critical parameters for SIMPLER, we determined experimentally the axial localization precision (σ_z) and compared it to the theoretical predictions (equation 3). From the same SIMPLER – DNA-PAINT experiment ($\hat{N}_0 = 51,000$), single-molecule emission events longer than 5 camera frames were selected to compute an experimental measure of σ_z . In this way, after filtering out the first and last frames (Figure 2a), at least three independent measurements of \hat{N} , and their corresponding estimations of \hat{z} , were available for every single molecule. Figure 3c shows the obtained distributions of experimental σ_z (extracted from 366 single-molecule traces) grouped for different ranges of z.”

In addition, at the end of that discussion, we have added the following sentence to make it absolutely clear that SIMPLER does not require any priors about the variances of any parameter:

“We remark though that this analysis is only performed to validate the applicability of equation 3. It is not necessary for the application of SIMPLER, as its implementation does not require any prior knowledge about the variances of the parameters involved.”

And in the Method section:

“Experimental measures of the z-localization precision (σ_z) were determined from the variance of z-localizations of the same molecule. Values of σ_z were registered for multiple molecules located at different z-positions as described in the main text (Figures 3e and 4d).”

>>> This comment is adequately addressed.

***Comment (3-6):** 6- I do not understand the results in Figure 1e which is supported by Supplementary figure 1a. From my reading of the text, the exact solution represents the product of the red line CF in figure 1d and the evanescent wave approximation (the blue line in figure 1e). When normalized at $z=0$, shouldn't the exact solution always display a steeper decline and a lower amplitude than the evanescent wave approximation at all values $z>0$?*

Response (3-6): We thank Reviewer #3 for this comment. It is just a confusion. The exponential curve shown in Figure 1e is not the evanescent excitation field, but a fit to the exact solution.

$I(z)$ is the evanescent excitation field (Figure 1b). The detected fluorescence signal $F(z)$ will be proportional to the product of the excitation field and the collected fluorescence: $F(z) = I(z) \times CF_{avg}(z)$. This is the exact solution, as it contains the numerically calculated collected fluorescence. It turns out that the exact solution (red curve in Figure 1e) is very well represented by an exponential function (blue curve in figure 1e). This comment made us note that the blue curve displayed in Figure 1e was mistakenly labeled as $I(z)$. We have now corrected it.

>>> This comment is sufficiently addressed.

***Comment (3-7):** 7- I am unconvinced that the uncertainty for the $N0$ value is of minor importance as indicated in the text. Could the authors elaborate and help clarify their argument for this conclusion? I understand from the Figure 2c and supplementary figure 2 that choosing a value of $N0$ that does not correspond to $z=0$ will lead to an offset in the position of the other molecules. But shouldn't this be plotted as an uncertainty in the position of the molecules rather than an offset? The histogram in figure 2b should provide σ_{N0} , correct?*

Response (3-7): Reviewer #2 also raised this point in Comment 2-4. Evidently, we were not clear on this passage of our manuscript. We refer Reviewer #3 to our Response 2-4. We believe that, with the improved explanation and new Supplementary Figures 1, 5, and 6, we have cleared all doubts about the robustness of SIMPLER and the influence of each parameter in its end performance.

>>> I appreciate that the authors ran numerous simulations to show the robustness of the method, but I still don't understand the reasoning of why the uncertainty of the reference point (N_0 and subsequently σ_{N_0}) is not accounted in the plotted images in SIMPLER. Since the N_0 value is very important in these analyses, it will be of benefit to clearly define its role and educate all of us who may wish to use this method. Using the axial position calculated when imaging single molecules over multiple frames is probably a reasonable way to estimate σ_Z for molecules in the sample, but to calculate those positions, the authors use equation 2 and insert the mean N_0 photons obtained from molecules deposited on the coverslip. Since the N_0 is a mean value, there is an uncertainty in this value.

Should not the axial position uncertainty of the molecules of interest be a combination of the uncertainties of both the reference molecules and the experimental molecules?

I don't know the proper calculation of the variance of the ratio of two variables or I would offer it here. However, I am sure the authors have considered this and rejected it as unnecessary. I would simply like them to clarify why this uncertainty should not be reflected in the final plot of the molecules.

The simulations showing that the molecules are offset when the wrong N_0 value is used are helpful. But in general, we will not know if we are using the wrong N_0 and thus we will calculate the wrong positions for the molecules. I think where the authors and I differ on this point is that I consider this to be additional uncertainty to the molecule position, whereas the authors consider it simply an offset of the positions of all molecules and it doesn't matter because the axial positions between the molecules remains the same. Please correct me if I am misinterpreting the authors' view on this point. I'm simply trying to summarize the disagreement so the authors can correct my misunderstanding.

My guess is that the uncertainty in the reference molecule positions will not add much to the uncertainty of the experimental molecule positions even if it is included. But since this technique may be widely adopted due to its relatively straightforward implementation and the authors' do emphasize vast improvement over existing techniques, SIMPLER is likely to be highly scrutinized after publication by readers as well as the authors of those publications.

Comment (3-8): 8- The authors indicate that the lower bound for the uncertainty might be estimated from the square root of the number photons. Figure 2b should offer an adequate test for this approximation. How well does the data in figure 2b approximate this theoretical lower bound? If the approximation is insufficient, the authors should use the uncertainty of the reference molecules located at $z=0$ as the practical lower bound.

Response (3-8): We do indicate $\sigma_N = \sqrt{N}$ is the theoretical lower bound. But we also say that "in real life experiments, other factors may enlarge this value. For example, the variance introduced by EM amplification in EM-CCD cameras used in SMLM can lead to errors in photon counts that are a factor of 2 larger than Poisson statistics⁵⁴."

In the revised version of the manuscript, we have improved this paragraph following a suggestion by Reviewer #1 (Comment 1-7), that micro-blinking could also contribute to enlarge σ_N . The new paragraph reads as follows:

“We note, however, that in real-life experiments, other factors may enlarge this value. For example, the variance introduced by EM amplification in EM-CCD cameras used in SMLM can lead to errors in photon counts that are a factor of 2 larger than Poisson statistics⁵⁴. Also, the presence of a faster blinking process occurring with rates comparable to the camera frame rate may increase the variability of N.”

The best experimental test for σ_N is to analyze it for individual molecules, as we do for example in Figures 3d and 3e, and 4d. The histogram of photon counts at $z = 0$ shown in Figure 2b is not a good test for σ_N because it corresponds to molecules distributed over a very large area of $16 \times 16 \mu\text{m}^2$. The width of this distribution is enlarged by other factors, such as imperfect corrections of the excitation intensity and even the size of the antibodies.

>>> Calculating the variance on a molecule by molecule basis seems reasonable. However, $z=0$ should provide the most precise localizations, especially since it provides the critical reference point for converting photons into axial position. Moreover, I would have expected these data to reproduce the analyses on individual molecules taken at multiple z positions as discussed in response 3-13 below. Supplementary figure 3 from the previous submission showed the field of view dependent distribution the authors mention above. I can't seem to find this figure with the new submission, but I think the authors should include it to help future SIMPLER users.

The authors mention in the text that the local intensities are in figure 2b are corrected but presumably these do not account for all of the variance and produce average N_0 sigma of $22 \cdot \sqrt{N}$ (from response 3-9). Some of the figures such as figure 3 and 4 appear to be from large areas (maybe $75\text{-}100 \mu\text{m}^2$) and the correction seemed to work well on them. Did the fields of reference molecules used for those data sets display N_0 sigma of $5 \cdot \sqrt{N}$ similar to the experimental molecules? I think the authors should include such information. It may be a helpful part of the protocol for future users of SIMPLER to check their field dependent variances.

I'm sure this will be instrument dependent, but it may be helpful for the readers to know the approximate cut-off for the field size used in the datasets shown in this study?

Comment (3-9): 9- From analyzing single molecule traces, the authors conclude that the photon uncertainty for molecules $z > 0$ can be estimated from $5 \cdot \sqrt{N}$. Does this hold for N_0 as well?

Response (3-9): We find that in our experiments, the average value of $\sigma_{\hat{N}}$ for single molecules, independently of their z coordinate, is of about $5\sqrt{N}$. This would hold for single molecules located at $z = 0$. However, the distribution of \hat{N}_0 from an ensemble of single-molecule distributed over a large field of view presents a larger variance, for the reasons explained in our previous answer (Response 3-8). In fact,

the histogram of \widehat{N}_0 that we show in Figure 2b is an example of this. It presents a mean value of 51,000 counts and a standard deviation of 10%. That is a standard deviation of 5,100 counts, $\sim 22 \times \sqrt{N}$.

>>> **The previous submission Supplementary figure 3 showed three different area sizes with σ_z decreasing with field size. It would be helpful to replot these data using photons per frame as was done with Figure 2b. This should show readers how to determine the field size where the reference molecule photon uncertainty is approximated well by $5 \times \sqrt{N}$ similar to the experimental molecules.**

Comment (3-10): 10- Some of the simulations (such as Figure 1g) indicate the axial uncertainty approaches zero as the axial position approaches $z=0$. Given that the authors find in figure 3 that $5 \times \sqrt{N}$ seems to offer a good approximation for axial uncertainty, it is unclear how it can approach zero even at $z=0$. Can the authors clarify or explain this?

Response (3-10): There seems to be a confusion. The value of $5\sqrt{\widehat{N}}$ is a good approximation for $\sigma_{\widehat{N}}$, not for σ_z .

The curves of σ_z vs. z shown in Figure 1g are representations of equation (3) for different values of the N_0 and σ_{d_F} . We have modified the caption of Figure 1g to make this clearer:

“(g) Theoretical lower bound for the axial localization precision of SIMPLER (equation 3) for different sets of N_0 and σ_{d_F} with $d_F = 87.5$ nm and $\alpha_F = 0.93$.”

Therefore, although very small at $z = 0$, σ_z can never take the value of zero (equation 3).

Nonetheless, in response to Comment (2-3) by Reviewer #2, we have improved Figure 1g for clarity in the new version of the manuscript (please see Response 2-3).

>>> **Apologies, my mistake. The simulations in Fig. 1g were performed with the theoretical lower bound of $\sqrt{\text{photons}}$ as opposed to $5 \times \sqrt{N}$.**

Comment (3-11): 11- The authors simulate the expected uncertainty for many of the components in equation 3 used to determine the axial positions (Fig. 1g, Fig. 2c, Supp. Fig 1, Supp. Fig. 2). Generally, the components of interest in each figure introduce <10nm uncertainty over the axial range, but it is unclear what uncertainties are assumed for the other components when making these simulations. It will be helpful if the authors include this information in the legend for each figure?

Response (3-11): In all cases where we show calculated σ_z curves (Fig. 1g, Fig. 3e, and Fig. 4d) we did it using equation (3), which expresses σ_z as a function of $\sigma_{\widehat{N}}$ and σ_{d_F} , and neglects $\sigma_{\widehat{N}_0}$ and σ_{α_F} . In response

to Comments (1-6) and (2-4) of the other Reviewers, we have included new Supplementary Figures 1, 5, and 6 to illustrate the influence of the involved parameters in the determination of axial positions.

We remark again that the performance of SIMPLER does not depend on any theoretical estimation of the variances of the involved parameters. We compute the theoretical lower bound through equation 3 and compare it to the experimentally determined σ_z (Figs. 3e and 4d) just to gain and provide insight into the critical parameters and sources of error. We have added the following sentences to make this clearer in the new version of the manuscript:

“We remark though that this analysis is only performed to validate the applicability of equation 3. It is not necessary for the application of SIMPLER, as its implementation does not require any prior knowledge about the variances of the parameters involved.”

>>> Actually, I think my question was more basic than the authors are interpreting. The simulations were made using equations 2 and 3 and these have inputs of df , N , and N_0 . I don't think the authors should have to simulate the entire matrix of different conditions, but they actually did several in Supplemental Figure 5. My suggestion was for the simulations, it would be beneficial to the reader to know what values of df , N , and N_0 were used. These variables have uncertainties which will be additional to the mis-localizations shown in Supplemental Figure 1 and we can get an idea of the final uncertainties in the molecule localizations. Supplemental Figure 5 adequately addresses this comment.

Comment (3-12): 12- The parameters df and af are derived from a fit of $I(z) \times CF(z)$, both of which are calculated based on instrument and sample parameters. The use of just these calculated values is worrisome. It would be more compelling if the authors offered experimental determinations for the values using their imaging systems to compare with the simulated values.

Response (3-12): We thank Reviewer #3 for this Comment. While it would be nice to measure $I(z)$ and $CF(z)$ independently, such measurements are not trivial and would imply experimental conditions different from the real biological samples, which in turn would call for approximations and calculations to adjust the measurement results to the new conditions. In summary, that is unfortunately not a viable approach. That said, we would like to remark that the end performance of SIMPLER is also a way to validate the values of d_F and α_F . Using structures of known and fixed geometry as standards, such as the microtubule cross-sections or the nuclear pore complex, it is straightforward to validate the values of d_F and α_F (and of N_0). Any mismatch in those values leads to distinctive distortions in the images, as summarized in Supplementary Table 2 and explained in more detail in the new Supplementary Figures 1, 5, and 6; please see Response 2-4.

In addition, in the new version of the manuscript, we provide a Supplementary Software to apply SIMPLER from a list of localizations. The software includes an intuitive graphical user interface and a specific module to test and adjust the values of θ_i , α and N_0 , using the retrieved SIMPLER image as feedback (an example of the output of this module is illustrated in the new Supplementary Figure 7).

We also mention this in the revised version of the main manuscript text:

“To ease the implementation of SIMPLER to the wide-imaging community, we also make available a Supplementary Software with an intuitive graphical user interface (GUI) that directly outputs z-position from 2D SMLM analysis lists (Supplementary Software 1). The software computes $F(z) = I(z) \times CF_{avg}(z)$ and provides d_F and α_F for each user input experimental conditions: NA ; λ_0 ; λ ; θ_i ; n_i ; n_s and α . Among other features, the software can correct photon counts for uneven illumination and automatically perform the frame filtering step. In addition, taking advantage of the distinctive effects of each of the calibration parameters θ_i , α and N_0 , (Supplementary Table 2, Supplementary Figures 1, 5 and 6) their values can be easily adjusted using 3D images of reference structures as feedback. The Supplementary Software includes a specific module for this purpose and, as illustrated in Supplementary Fig. 7, it is extremely useful to find the best estimate for a parameter that has been determined or estimated with low accuracy.”

>>> **This comment is sufficiently addressed.**

Comment (3-13): 13- It is unclear how the uncertainty for the TIRF depth, σ_{d_F} , and the uncertainty for the non-evanescent component, σ_{α_F} , are estimated in each experiment. Given they are required to calculate the axial position uncertainty, good estimates for these values are likely critical. The authors have included simulations for these which indicate they have little bearing on the axial position uncertainty. However, it is unclear if those simulated uncertainties reflect uncertainties in an experimental setting. Could the authors please elaborate on the estimation of these uncertainties?

Response (3-13): The uncertainties of σ_{d_F} and σ_{α_F} are not estimated for each experiment. That is not necessary to implement SIMPLER, nor is the a priori estimation of any variance of the other involved parameters.

What we do is to determine experimentally the value of σ_z from direct measurements, without any prior calculation, from the variance in multiple z-localizations of single-molecules located at different z-positions (Figures 3e and 4d). Then, we compare the experimentally determined values of σ_z vs. z to the calculated uncertainties through equation 3, using sensible ranges of σ_{d_F} and $\sigma_{\hat{N}}$ with the only purpose of gaining insight into the individual contributions to the final localization error. We remark this is not necessary for the application of SIMPLER. It is just a way to gain an understanding on the influence of the different parameters. In the revised version of the manuscript, this is done in much more detail. This issue was addressed also in our Response (1-6) to Reviewer #1 and Response (2-4) to Reviewer #2.

>>> **This clarification is satisfactory.**

Minor points

Comment (3-14):• Page 22; sentence 2. Typo.

Response (3-14): We thank the reviewer for pointing out this typo. We have corrected it in the current version of the manuscript.

***Comment (3-15):**• For one of the super-resolution microscopes, a sCMOS camera was used in the single molecule localization experiments, but it was unclear from the methods if the corrections for the pixel dependent noise common in these cameras was applied. Given the noise characteristics for these cameras, this correction is usually required for single molecule imaging. If they authors performed these corrections, please indicate this in the methods. If not, the authors may wish to revisit data from images produced by this instrument.*

Response (3-15): We thank Reviewer #3 for noticing this missing information. We used the intrinsic hot pixel correction of the sCMOS ORCA Flash 4.0 V3, Hamamatsu camera. This feature identifies erroneous hot pixels by comparing a pixel to its nearest neighbours. We have now clarified this in the methods section of the manuscript.

>>>I think the hot pixels are a separate issue. I was referring to the potential non-uniform pixel characteristics of the sCMOS sensor. There are several variations since this paper (Nature Methods (2013) 10:653-658). The presence or absence of such a calibration/correction may have little effect on the results presented here, but it is something authors should include if they used it.

Response to reviewers NCOMMS-20-13404-T

We thank again the reviewers for the careful reading and constructive comments. Below, we give a point by point response to the comments, and we note when modifications were done to the paper. To simplify response readout, we have not included Reviewer's 3 comments to responses 3-1, 3-4 to 3-6 and 3-10 to 3-14 as the reviewer has found that our previous responses adequately addressed her/his concerns.

Reviewer's comments in *italic*

Reviewer #1 (Remarks to the Author):

Comment (1-1): *I am satisfied with the response and believe the revised MS is greatly improved. I especially liked the software provision and came across 2 small bugs for which I include diffs below.*

While I like the software provision and example data a lot I would request that example localisation data covering slightly larger regions of interest are provided, too. A data volume 10x what is provided right now would be entirely acceptable for readers and provide a better overview of the data that can be achieved.

```
--- export_function.m~ 2020-07-11 22:52:00.000000000 +0100
+++ export_function.m 2020-08-16 22:50:13.000000000 +0100
@@ -51,13 +51,13 @@
simpler_output(:,3)>yl(1) & simpler_output(:,3)<yl(2));
end
x1 = simpler_output(c1,1);
- y1 = simpler_output(c1,1);
+ y1 = simpler_output(c1,2);
z = simpler_output(c1,3);
exportation_file_xyz(x1,y1,z,filename_wformat,handles);
elseif rz_xyz == 4
c2 = 1:size(simpler_output,1);
x1 = simpler_output(c2,1);
- y1 = simpler_output(c2,1);
+ y1 = simpler_output(c2,2);
z = simpler_output(c2,3);
exportation_file_xyz(x1,y1,z,filename_wformat,handles);
end
```

Response (1-1): We would like to thank specially Reviewer 1 for testing the supplementary software. We have now corrected the bugs reported by the reviewer and added comments in key passages of the software to facilitate the implementation and modification by any new user. Also, following the suggestion by Reviewer 1, we provide an extended data set that include a larger field of view of membrane-associated periodic skeleton of axons imaged in hippocampal neurons.

Reviewer #2 (Remarks to the Author):

Comment (2-1): *The authors have candidly addressed all the concerns of the reviewers, in particular mine. They have included new supplementary data to support some of their initial unclear statements,*

improved the explanation regarding some figures and included some new sentences that address my initial concerns. I have no hesitation to recommend this paper for Nature Comm.

Response (2-1): We thank the Reviewer 2 very much for the positive appraisal of our revised submission, the reviewer comments have surely allowed us to improve our original submission.

Reviewer #3 (Remarks to the Author):

Szalai et al. have addressed most of the comments satisfactorily. My additional comments to the rebuttal are placed in bold type in line below. I am still positive about the paper since it represents a great addition to the super-resolution imaging toolbox. I have just a points on which I am confident the authors can offer clarification or convince me of my misunderstanding. All of points concern the axial localization uncertainty and since it is arguably the most important factor for a localization method, it may benefit the authors to address the simplest and perhaps even naïve questions at this stage rather than after publication.

Comment (3-2-v2): *Thanks for this clarification to address my misunderstanding concerning the plots. It raises the issue of plotting localizations instead of molecules. Although it is more or less standard practice in the field, I do not consider this to be a good approach. Plotting at the lower sigma suggests the molecules are better localized than they actually are localized. In addition, under-sampling is often a problem with localization data and can lead to limitations on reaching Nyquist criteria. Thus, plotting the same molecule at slightly different positions has the effect of “connecting the dots” which are often the same dot. I recognize that localization data can be plotted in numerous ways and I’ll respect the authors’ decision to plot their data in this manner and simply inform the reader how it was performed. To that end, inclusion of the statements above will suffice for that purpose.*

I urge the authors to reconsider this approach for plotting data in future SIMPLER publications. Moreover, this particular study has an additional consideration regarding the plotting of molecules. For the lateral localizations, the uncertainty is largely determined for each one based on the number of photons, background, etc. For the axial localizations in the work, the uncertainty is determined by calculating the axial position of the same molecule in multiple frames. Thus, the determined uncertainty is for the mean position of the molecule rather than the individual localizations. It is not clear that the uncertainty in the mean axial position is applicable to the uncertainty in the individual axial localization positions. If the authors could provide some clarification on this point, it would be helpful for me and would likely be helpful for future users of SIMPLER.

Response (3-2-v2): We thank the Reviewer 3 for the recommendation on changing our approach of plotting 3D data in future SIMPLER publications. We think this is a good idea, and in future versions of our software, we will consider adding a new functionality to allow users to choose how to plot their 3D data (plotting either individual localisations or merged localisations per detected molecule).

Regarding the concern about comparing the theoretical standard error of individual axial localisations to the experimental standard error obtained from the standard deviation of single molecule traces, we note that this is a common approach in 3D SMLM methods that relies on photometric measurements to infer axial position. See for example, DONALD (Nature Photonics, 9, 587-593, 2015) or TRABI (Nature Methods, 14, 41-44, 2017), where axial localisation precision is also reported as the standard deviation of multiple determinations of the z-position of dye molecules located at the same z depth.

Comment (3-3-v2): *Including rationale for the 20nm cutoff for future users of SIMPLER to set this parameter will be helpful. Otherwise, these clarifications are satisfactory.*

Response (3-3-v2): Reviewer 3 is right that a rationale for this cut-off should be included. In order to assign consecutive single-molecule localisations to the same fluorophore it is common practice to define a cut-off value in the order of 3-5 times $\sigma_{x,y}$, which in our case is ~ 20 nm (i.e. $\sigma_{x,y}$ is $\sim 3-4$ nm for DNA-PAINT and $\sim 6-10$ nm for dSTORM data). We have modified the following passage in the manuscript to clarify our choice for the 20 nm cut-off value:

“To ensure the molecule emitted during the whole exposure time, localizations were kept as valid only in the case that other localizations, reasonably attributed to the same fluorophore (within 3-5 times $\sigma_{x,y}$, i.e. 20 nm), were detected in the previous and subsequent frames (Figure 2a).”

Furthermore, we note that users of SIMPLER can modify this value in the Supplementary Software.

Comment (3-7-v2): *I appreciate that the authors ran numerous simulations to show the robustness of the method, but I still don't understand the reasoning of why the uncertainty of the reference point (N_0 and subsequently σ_{z0}) is not accounted in the plotted images in SIMPLER. Since the N_0 value is very important in these analyses, it will be of benefit to clearly define its role and educate all of us who may wish to use this method. Using the axial position calculated when imaging single molecules over multiple frames is probably a reasonable way to estimate σ_{z0} for molecules in the sample, but to calculate those positions, the authors use equation 2 and insert the mean N_0 photons obtained from molecules deposited on the coverslip. Since the N_0 is a mean value, there is an uncertainty in this value. Should not the axial position uncertainty of the molecules of interest be a combination of the uncertainties of both the reference molecules and the experimental molecules?*

I don't know the proper calculation of the variance of the ratio of two variables or I would offer it here. However, I am sure the authors have considered this and rejected it as unnecessary. I would simply like them to clarify why this uncertainty should not be reflected in the final plot of the molecules. The simulations showing that the molecules are offset when the wrong N_0 value is used are helpful. But in general, we will not know if we are using the wrong N_0 and thus we will calculate the wrong positions for the molecules. I think where the authors and I differ on this point is that I consider this to be additional uncertainty to the molecule position, whereas the authors consider it simply an offset of the positions of all molecules and it doesn't matter because the axial positions between the molecules remains the same.

Please correct me if I am misinterpreting the authors' view on this point. I'm simply trying to summarize the disagreement so the authors can correct my misunderstanding. My guess is that the uncertainty in the reference molecule positions will not add much to the uncertainty of the experimental molecule positions even if it is included. But since this technique may be widely adopted due to its relatively straightforward implementation and the authors' do emphasize vast improvement over existing techniques, SIMPLER is likely to be highly scrutinized after publication by readers as well as the authors of those publications.

Response (3-7-v2): We think the concerns expressed in these three comments can be cleared out by reminding Reviewer 3 that, even though we count with an analytical expression for calculating the axial position uncertainty (Equation 3), we are not using it to determine the axial localization uncertainty for each single-molecule events. This being the first implementation of SIMPLER, we have opted to test the validity of the analytical expression by contrasting it to the experimental results (Figures 3e and 4d). The experimental measurement of the axial localization uncertainty was obtained from the variance in axial position of single molecules detected repeatedly over multiple frames.

The experimental axial localization uncertainty includes the contributions of all sources of variability (σ_N , σ_{d_F} , σ_{α_F} , and σ_{N_0}). What we find is that the experimentally determined uncertainty is well described by our analytical expression and reasonable ranges of uncertainty for the critical parameters σ_N and σ_{d_F} . Of course, equation 3 could be extended to include σ_{α_F} and σ_{N_0} , but for simplicity, we

decided to keep the analysis around σ_N and σ_{d_F} , and consider the effects of α_F and N_0 through the simulations provided in the supplementary information. Presumably in the near future, as we gain experience and confidence in using SIMPLER, it will be possible to use the analytical expressions to assign reliably axial uncertainties to individual single-molecule events, based on prior knowledge of experimental variables.

Comment (3-8-v2): *Calculating the variance on a molecule by molecule basis seems reasonable. However, $z=0$ should provide the most precise localizations, especially since it provides the critical reference point for converting photons into axial position. Moreover, I would have expected these data to reproduce the analyses on individual molecules taken at multiple z positions as discussed in response 3-13 below. Supplementary figure 3 from the previous submission showed the field of view dependent distribution the authors mention above. I can't seem to find this figure with the new submission, but I think the authors should include it to help future SIMPLER users. The authors mention in the text that the local intensities are in figure 2b are corrected but presumably these do not account for all of the variance and produce average N_0 sigma of $22 \cdot \sqrt{N}$ (from response 3-9). Some of the figures such as figure 3 and 4 appear to be from large areas (maybe 75-100 μm^2) and the correction seemed to work well on them. Did the fields of reference molecules used for those data sets display N_0 sigma of $5 \cdot \sqrt{N}$ similar to the experimental molecules? I think the authors should include such information. It may be a helpful part of the protocol for future users of SIMPLER to check their field dependent variances. I'm sure this will be instrument dependent, but it may be helpful for the readers to know the approximate cut-off for the field size used in the datasets shown in this study?*

Comment (3-9-v2): *The previous submission Supplementary figure 3 showed three different area sizes with sigmaZ decreasing with field size. It would be helpful to replot these data using photons per frame as was done with Figure 2b. This should show readers how to determine the field size where the reference molecule photon uncertainty is approximated well by $5 \cdot \sqrt{N}$ similar to the experimental molecules.*

Response (3-8-v2) and (3-9-v2): The variance in photon counts for any single-molecule was found to be of about $5 \cdot \sqrt{N}$ throughout the axial range of SIMPLER (Figure 3d). The variance of N_0 shown in Figure 2b ($22 \cdot \sqrt{N}$) is significantly larger than $5 \cdot \sqrt{N}$ because that distribution includes molecules spread over 256 μm^2 and, consequently, involves a large variety of correction factors for each pixel and Fab orientations, which in turn may translate in z -positions of the fluorophores ranging from 0 to ~5 nm.

Following the suggestion made by Reviewer 3, in the revised version of our supplementary information, we include again the original Supplementary Figure 3, as new Supplementary Figure 12. Nonetheless, we would like to remark that there is no need to apply cut-offs in the field of view. The new Supplementary Figure 12 shows that over a field of view of 25 μm^2 , the axial uncertainty is of just 7 nm (practically the size of the Fab fragments). As the field of view increases, the overall axial variability increases, reaching 16 nm in an area of 256 μm^2 . This should be thought of as a variable axial absolute position over that large area. For example, the absolute axial position of a microtubule extending over the 256 μm^2 will be uncertain to 16 nm, but its cross-section at any given point will be accurately reconstructed in 3D with a localization accuracy as reported in any of the examples.

Finally, Reviewer 3 is concerned that the goodness of the correction may be instrument dependent. We believe that with the explanations made, he/she will now have a clear understanding of the influence of the intensity correction. Nevertheless, we would like to stress that our work includes reliable 3D SIMPLER reconstructions obtained from 2D SMLM data acquired in two completely different set-ups, in different labs, and by different users.

Comment (3-15-v2): *I think the hot pixels are a separate issue. I was referring to the potential non-uniform pixel characteristics of the sCMOS sensor. There are several variations since this paper*

(Nature Methods (2013) 10:653-658). The presence or absence of such a calibration/correction may have little effect on the results presented here, but it is something authors should include if they used it.

Response (3-15-v2): The reviewer is right to point out that pixel-dependent noise calibration for sCMOS cameras have been reported to improve localisation precision (Nature Methods (2013) 10:653-658). However, similarly to others in the field (see for example, Nature Protocols, 12, 1198-1228 (2017)) we have not accounted for this correction in our data analysis pipeline. We will take this into account in future experiments, and test if this correction provides any further improvement for SIMPLER.

REVIEWER COMMENTS

Reviewer #3 (Remarks to the Author):

The authors have sufficiently addressed most of my concerns, but my major criticism concerns the uncertainty of the axial localizations. The authors and I seem to have a disagreement in how to treat uncertainty in the reference molecule data and, by extension, the experimental data. Therefore, I do not think the authors adequately addressed these points and thus I am not completely convinced of SIMPLER's capabilities. Since this is the most critical part of the SIMPLER approach, I cannot endorse publication. However, there also seems to be some miscommunication (see Response 3-7-v2) between the authors and myself which has contributed to our inability to satisfy this disagreement. Even with better communication, I suspect the authors and I will continue to disagree over this point. I base this on a comment from the authors in Response (3-8-v2 and 3-9-v2) below. There is probably little reason to extend my review of this manuscript but I also think it will be unfair to adamantly oppose publication based on this disagreement. Given this work will likely be published soon, the authors should consider that it might be helpful for readers if they clarify their reasoning in the points below and include them within discussions in the manuscript.

Response (3-7-v2): We think the concerns expressed in these three comments can be cleared out by reminding Reviewer 3 that, even though we count with an analytical expression for calculating the axial position uncertainty (Equation 3), we are not using it to determine the axial localization uncertainty for each single-molecule events. This being the first implementation of SIMPLER, we have opted to test the validity of the analytical expression by contrasting it to the experimental results

(Figures 3e and 4d). The experimental measurement of the axial localization uncertainty was obtained

from the variance in axial position of single molecules detected repeatedly over multiple frames.

Response to Response (3-7-v2): This response seems to be a miscommunication. I did not refer to equation 3 but rather equation 2 which converts photons in each spot into axial position based on the mean photon number at $z=0$. Unless I am misunderstanding the approach, molecules lasting >5 frames excluding the first and last frames were used. The number of photons per spot per frame was used to determine the axial position using equation 2. From the variance in the axial positions for the same molecule, the uncertainty in the axial localization is derived. If this is not the way the data such as those in Figure 3d were derived, perhaps the authors can explicitly indicate in the text or the figure legend how the axial positions and their subsequent uncertainties were determined.

However, if equation 2 is used in these determinations, this comes back to my previous concern.

Calculation of the axial position requires the mean photon number at $z=0$. Since it is a mean value, there will be an uncertainty associated with it. I consider that this error will propagate into the molecule axial localization uncertainty. From supplementary figures 1d and 6, the authors view uncertainties in $N_{z=0}$ as simply an offset in the entire dataset which will shift the structures axially. I am not convinced this is the proper treatment of the uncertainty in the reference position ($z=0$). In the imaging methods measuring the lateral distance between two points of which I am aware, the uncertainty of that distance is a combination of the uncertainty in the localization of both points. Although SIMPLER is not exactly the same as the lateral methods, it is doing something similar, calculating the distance between the $z=0$ reference point molecules and experimental molecule axial positions based on their relative photon counts.

Of course, if the authors are not using equation 2 to determine the axial positions and hence the localization uncertainty, then I am not understanding the SIMPLER concept of how the number of photons is converted into an axial position and the authors and editors should feel free to ignore my comments.

Response (3-8-v2) and (3-9-v2): The variance in photon counts for any single-molecule was found to be of about $5\sqrt{N}$ throughout the axial range of SIMPLER (Figure 3d). The variance of σ shown in Figure 2b ($22\sqrt{N}$) is significantly larger than $5\sqrt{N}$ because that distribution includes molecules spread over $256\ \mu\text{m}^2$ and, consequently, involves a large variety of correction factors for each pixel and Fab orientations, which in turn may translate in z-positions of the fluorophores ranging from 0 to ~ 5 nm.

Following the suggestion made by Reviewer 3, in the revised version of our supplementary information, we include again the original Supplementary Figure 3, as new Supplementary Figure 12. Nonetheless, we would like to remark that there is no need to apply cut-offs in the field of view. The new Supplementary Figure 12 shows that over a field of view of $25\ \mu\text{m}^2$, the axial uncertainty is of just 7 nm (practically the size of the Fab fragments). As the field of view increases, the overall axial variability increases, reaching 16 nm in an area of $256\ \mu\text{m}^2$. This should be thought of as a variable axial absolute position over that large area. For example, the absolute axial position of a microtubule extending over the $256\ \mu\text{m}^2$ will be uncertain to 16 nm, but its cross-section at any given point will be

accurately reconstructed in 3D with a localization accuracy as reported in any of the examples. Finally, Reviewer 3 is concerned that the goodness of the correction may be instrument dependent. We believe that with the explanations made, he/she will now have a clear understanding of the influence of the intensity correction. Nevertheless, we would like to stress that our work includes reliable 3D SIMPLER reconstructions obtained from 2D SMLM data acquired in two completely different set-ups, in different labs, and by different users.

Response to Response (3-8-v2) and (3-9-v2): This is related to the previous point, namely concerning the uncertainty in the axial localizations. The authors do not seem to consider the uncertainty in the reference point ($z=0$) to be of much concern, whereas I do consider it problematic. With the authors' view, they are correct in ignoring the field of view dependence on the axial uncertainty and should conclude that the fields of view do not need to be matched for comparing the number of photons in their experimental samples with the data from coverslip attached reference molecules. In my view, the uncertainty is obviously field of view dependent since the localization in the reference data ranges from $0\pm 7\text{nm}$ at the smaller fields of view to $0\pm 16\text{nm}$ at the larger fields of view. Actually, the authors and I agree on this point based on the following from their statement above.

"For example, the absolute axial position of a microtubule extending over the $256\ \mu\text{m}^2$ will be uncertain to 16 nm, but its cross-section at any given point will be accurately reconstructed in 3D with a localization accuracy as reported in any of the examples."

I think our disagreement is over how this should be interpreted. If I'm understanding their argument, the authors consider the molecules should be plotted with the small Gaussians distributions ($2\text{-}6\text{nm}$) which may or may not be simply offset from their real axial position by 16 nm. However, I consider this 16 nm to be part of the uncertainty and that the molecules should be plotted to reflect it. The authors' arguments in the manuscript and rebuttals have not convinced me otherwise.

My reasoning follows that since the reference molecules are located at $z=0$, they should produce the brightest signals and thus show the most precision in their axial localization. I would not expect experimental molecule uncertainties to be better than these values and I would actually expect them to be worse since I also think the uncertainty in $z=0$ data should also be accounted in the experimental molecule localization uncertainties.

Regarding the field of view issue. Since the uncertainties of the $z=0$ data in the small fields ($\pm 7\text{nm}$) of view more closely match with the arguments of having $<10\text{nm}$ uncertainty through the axial range, I thought the authors had made comparisons between similar sized fields of view for the reference data. The authors' response indicates this was not done. Again, I think this arises from our basic disagreement as mentioned above.

Response to reviewers
NCOMMS-20-13404B

Reviewer's comments in *italic*
Authors' responses intercalated in red.

Reviewer #3 (Remarks to the Author):

The authors have sufficiently addressed most of my concerns, but my major criticism concerns the uncertainty of the axial localizations. The authors and I seem to have a disagreement in how to treat uncertainty in the reference molecule data and, by extension, the experimental data. Therefore, I do not think the authors adequately addressed these points and thus I am not completely convinced of SIMPLER's capabilities. Since this is the most critical part of the SIMPLER approach, I cannot endorse publication. However, there also seems to be some miscommunication (see Response 3-7-v2) between the authors and myself which has contributed to our inability to satisfy this disagreement. Even with better communication, I suspect the authors and I will continue to disagree over this point. I base this on a comment from the authors in Response (3-8-v2 and 3-9-v2) below. There is probably little reason to extend my review of this manuscript but I also think it will be unfair to adamantly oppose publication based on this disagreement. Given this work will likely be published soon, the authors should consider that it might be helpful for readers if they clarify their reasoning in the points below and include them within discussions in the manuscript.

Response to Response (3-7-v2-a): *This response seems to be a miscommunication. I did not refer to equation 3 but rather equation 2 which converts photons in each spot into axial position based on the mean photon number at $z=0$. Unless I am misunderstanding the approach, molecules lasting >5 frames excluding the first and last frames were used. The number of photons per spot per frame was used to determine the axial position using equation 2.*

From the variance in the axial positions for the same molecule, the uncertainty in the axial localization is derived. If this is not the way the data such as those in Figure 3d were derived, perhaps the authors can explicitly indicate in the text or the figure legend how the axial positions and their subsequent uncertainties were determined.

Response to Reviewer (3-7-v2-a): The description made by Referee 3 of the way we derive axial positions and the corresponding uncertainty using SM traces lasting >5 frames is correct.

Response to Response (3-7-v2-b): *However, if equation 2 is used in these determinations, this comes back to my previous concern. Calculation of the axial position requires the mean photon*

number at $z=0$. Since it is a mean value, there will be an uncertainty associated with it. I consider that this error will propagate into the molecule axial localization uncertainty.

From supplementary figures 1d and 6, the authors view uncertainties in $N_{z=0}$ as simply an offset in the entire dataset which will shift the structures axially. I am not convinced this is the proper treatment of the uncertainty in the reference position ($z=0$). In the imaging methods measuring the lateral distance between two points of which I am aware, the uncertainty of that distance is a combination of the uncertainty in the localization of both points. Although SIMPLER is not exactly the same as the lateral methods, it is doing something similar, calculating the distance between the $z=0$ reference point molecules and experimental molecule axial positions based on their relative photon counts.

Of course, if the authors are not using equation 2 to determine the axial positions and hence the localization uncertainty, then I am not understanding the SIMPLER concept of how the number of photons is converted into an axial position and the authors and editors should feel free to ignore my comments.

Response to Response (3-8-v2) and (3-9-v2): This is related to the previous point, namely concerning the uncertainty in the axial localizations. The authors do not seem to consider the uncertainty in the reference point ($z=0$) to be of much concern, whereas I do consider it problematic. With the authors' view, they are correct in ignoring the field of view dependence on the axial uncertainty and should conclude that the fields of view do not need to be matched for comparing the number of photons in their experimental samples with the data from coverslip attached reference molecules. In my view, the uncertainty is obviously field of view dependent since the localization in the reference data ranges from $0\pm 7\text{nm}$ at the smaller fields of view to $0\pm 16\text{nm}$ at the larger fields of view. Actually, the authors and I agree on this point based on the following from their statement above.

“For example, the absolute axial position of a microtubule extending over the $256\ \mu\text{m}^2$ will be uncertain to 16 nm, but its cross-section at any given point will be accurately reconstructed in 3D with a localization accuracy as reported in any of the examples.”

I think our disagreement is over how this should be interpreted. If I'm understanding their argument, the authors consider the molecules should be plotted with the small Gaussians distributions (2-6nm) which may or may not be simply offset from their real axial position by 16 nm. However, I consider this 16 nm to be part of the uncertainty and that the molecules should be plotted to reflect it. The authors' arguments in the manuscript and rebuttals have not convinced me otherwise.

My reasoning follows that since the reference molecules are located at $z=0$, they should produce the brightest signals and thus show the most precision in their axial localization. I would not expect experimental molecule uncertainties to be better than these values and I would actually expect them to be worse since I also think the uncertainty in $z=0$ data should also be accounted in the experimental molecule localization uncertainties.

Regarding the field of view issue. Since the uncertainties of the $z=0$ data in the small fields ($\pm 7\text{nm}$) of view more closely match with the arguments of having $<10\text{nm}$ uncertainty through the axial range, I thought the authors had made comparisons between similar sized fields of view for the reference data. The authors' response indicates this was not done. Again, I think this arises from our basic disagreement as mentioned above.

Response to Reviewer (3-7-v2-b), (3-8v2), and (3-9-v2): We believe that we can clear all remaining doubts to Referee 3 by explaining in more detail the influence of the variability of N_0 on the super-resolved 3D images.

Reviewer 3 is right in that the determination of the axial position requires knowledge of N_0 . N_0 is the number of photons that the fluorescent probe would give when placed at the interface. This parameter is defined by the experimental set-up (laser intensity, fluorophore, filter-set, camera, etc). Reviewer 3 is right that, in practice, the value of N_0 is field-of-view dependent. In fact, in practice, the exact value of N_0 may vary from one lateral position to another over the sample, because illumination is never perfectly uniform and can never be perfectly corrected.

Ideally, one would like to know the exact value of N_0 at each lateral position. However, performing such a measurement over an entire field of view of several squared micrometers is practically impossible. We determined the average value of N_0 over the field of view and used it for all localizations. Below, we explain the consequences of this procedure. But before, we would like to note that this is not an unusual situation in this type of measurements. For example, the widely used method of axial localization through astigmatic imaging also uses an average calibration curve obtained throughout the complete field of view, although the exact astigmatic behavior may also vary from one position to another of the sample.

We determined N_0 from a reference measurement of molecules spread over a coverslip. An example distribution of N_0 obtained in such a measurement for a field of view of $256\ \mu\text{m}^2$ is shown in Figure 2b. That distribution has a standard deviation of 10%.

If we interpret Reviewer 3 correctly, he/she considers that this 10% uncertainty found in N_0 over a large area should be propagated in the uncertainty of the axial position determined through SIMPLER of any molecule located in the field of view. This is not correct, because that 10% is not the uncertainty in the local value of N_0 . It is the variability of N_0 over a large area due to the variations of laser intensity over the large area. Local variations of N_0 are only expected due to temporal fluctuations of laser intensity at that position, which are minor and included in the experimentally determined variations of axial positions. We provide experimental proof of this. In Supplementary Figure 12, we show an example measurement of axial positions of Fab fragments adsorbed directly to the coverslip. These measurements were done using DNA-PAINT and a single value of N_0 . The distribution of axial positions obtained in the full FOV ($256\ \mu\text{m}^2$) has a standard

deviation of 16 nm. If a reduced field of view of 128 μm^2 is measured, the standard deviation reduces to 12 nm. For a region of 25 μm^2 it further reduces to 7 nm. For any given lateral position, the uncertainty in z will be even smaller. Indeed, for fixed lateral positions the axial uncertainty will be the one obtained from the single-molecule measurements reported in Figures 3e, namely 4 nm (median). That is the reason why SIMPLER delivers such clear cross-sections of microtubules.

Another way of seeing this is considering that the large area variability of N_0 of 10% corresponds to $22\sqrt{N}$, while the variability in N of single molecules at any axial position is $\sim 5\sqrt{N}$ (Figures 3d, 3e, and 4d). We remark that this experimentally determined variability of N already includes all sources of error, including local variations of N_0 due to temporal laser fluctuations at the molecule position.

In summary, the large area variability of N_0 does not compromise the local axial positioning precision achievable with SIMPLER. Nonetheless, using a single value of N_0 for the whole field of view does have an impact on the absolute axial positions. The variability in N_0 determines the uncertainty in absolute axial positioning when comparing structures from different regions. Next, we explain this in further detail.

Let us consider the distribution of N_0 shown in Figure 2b, obtained from molecules dispersed over a sample area of 256 μm^2 of a coverslip. The distribution is nearly normal with a standard deviation of about 10%. In Supplementary Figure 1d, we quantified the effect of using SIMPLER with wrong values of N_0 . An overestimation of N_0 by 10%, leads to a mislocalization $\Delta z = 8.5$ nm for molecules at $z = 0$. For molecules at $z = 150$ nm Δz increases to 11.0 nm. And for molecules at $z = 250$ nm $\Delta z = 18$ nm. A similar behavior is found for an underestimation of N_0 by 10%, but with negative mislocalizations. Therefore, using SIMPLER to locate molecules at z between 0 and 150 nm with a 10% wrong N_0 basically leads to an axial off-set of 10 nm (± 1 nm). For molecules between 150 and 250 nm the mislocalization increases with z and leads to small, probably undetectable, distortions. Let us take for example a structure spanning 100 nm in z , with its bottom located at $z = 150$ nm and its top at $z = 250$ nm. Molecules placed at the bottom will be mislocated by +11.0 nm. Molecules placed at the top will be mislocated by + 18 nm. Thus, the structure will appear to be of 108 nm instead of 100 nm. This is a difference of just 8 nm in 100 nm.

These examples are meant to show that locally, the effect of using a 10% wrong N_0 are negligible in terms of resolution and axial distortions. Now, if one aims to compare absolute axial positions of molecules throughout the complete field of view, then the large area variability of N_0 imposes a limit. We remark again, this is an issue for any method using an average calibration, not a local calibration. Nonetheless, the uncertainty in absolute axial positions are really small, 10 – 16 nm for areas of 100 to 256 μm^2 .

Finally, to make this clearer in the manuscript, we have extended the discussion and included a specific reference to Supplementary Figure 1d, which was missing (there was only a general reference to Supplementary Figure 1):

“Such a determination of \widehat{N}_0 is sufficiently accurate for SIMPLER, as shown in the performance examples of the next section where the 3D nanoscale organization of different biological structures are resolved. Specifically, we have quantified the effect of using SIMPLER with a wrong value of \widehat{N}_0 in Supplementary Figure 1d. It turns out that applying SIMPLER with the wrong \widehat{N}_0 produces mainly a localization off-set for molecules located at z between 0 and 150 nm. For molecules located at larger z , small axial distortions are introduced. For example, an overestimation of \widehat{N}_0 by 10% leads to a mislocalization $\Delta z = 8.5$ nm for molecules at $z = 0$. For molecules at $z = 150$ nm, Δz increases to 11.0 nm, and for molecules at $z = 250$ nm Δz is 18.1 nm. A similar behavior with negative mislocalizations is found for an underestimation of \widehat{N}_0 by 10%.”

“On the other hand, it should also be noticed that, while the axial localization precision at any given point within the field of view is well described by equation (3), if we compare the absolute axial position of different molecules over the entire field of view, the axial position variability will be higher. This is due to the position-dependent variability of \widehat{N}_0 throughout the field of view because the illumination is not perfectly uniform or corrected.”

REVIEWERS' COMMENTS

Reviewer #3 (Remarks to the Author):

The authors have addressed my criticisms.